

# Impact of urbanization on fine particulate matter concentrations over central Europe

Peter Huszar, Alvaro Patricio Prieto Perez, Lukáš Bartík, Jan Karlický, and Anahi Villalba-Pradas

Department of Atmospheric Physics, Faculty of Mathematics and Physics, Charles University, Prague, V Holešovičkách 2, 18000, Prague 8, Czech Republic

**Correspondence:** P. Huszar (peter.huszar@matfyz.cuni.cz)

**Abstract.** The rural-to-urban transformation (RUT) is the process of turning rural or natural land-surface into urban one which brings important modifications in the surface causing well know effects like the urban heat island (UHI), reduced wind-speeds, increased boundary layer heights and so on. Moreover, with concentrated human activities RUT introduces new emission source which greatly perturbs the local and regional air-pollution. Particulate matter (PM) is one of key pollutants responsible for deterioration of urban air-quality and is still a major issue in European cities with frequent exceedances of limit values. Here we introduce a regional chemistry-climate model (regional climate model RegCM coupled offline to chemistry transport model CAMx) study which quantifies how the process of RUT modified the PM concentrations over central Europe including the underlying controlling mechanisms that contribute to the final PM pollution. Apart from the two most studied ones, i) the urban emissions and ii) the urban canopy meteorological forcing (UCMF, i.e. the impact of modified meteorological conditions on air-quality) we analyze also two less studied contributors to the RUT's impact on air-quality: iii) the impact of modified dry-deposition velocities due to urbanized land-use and iv) the impact of modified biogenic emissions due to urbanization induced vegetation modifications and changes in meteorological conditions which affect these emissions. To calculate the magnitude of each of these RUT contributors, we perform a cascade of simulations were each contributor is added one-by-one to the reference state while focus is given on PM2.5 (particulate matter with diameter less then 2.5 μm). We also look at their primary and secondary components, namely primary elemental carbon (PEC), sulphates (PSO4), nitrates (PNO3), ammonium (PNH4) and secondary organic aerosol (SOA).

The validation using surface measurements showed a systematic negative bias for the total PM2.5 which is probably caused by underestimated organic aerosol and partly by the negative bias in sulphates and elemental carbon. For ammonium and nitrate, the underestimation is limited to the warm season while for winter, the model tends to overestimate their concentrations. However, in each case, the annual cycle is reasonably captured.

We evaluated the RUT impact on PM2.5 over an ensemble of 19 central European cities and found that the total impact of urbanization is about 2-3 and 1-1.5 $\mu$gm$^{-3}$ in winter and summer, respectively. This is mainly driven by the impact of emissions alone causing a slightly higher impact (1.5-3.5 and 1.2-2 $\mu$gm$^{-3}$ in winter and summer), while the effect of UCMF was a decrease at about 0.2-0.5 $\mu$gm$^{-3}$ (in both seasons) which was mainly controlled by enhanced vertical eddy-diffusion while increases were modelled over rural areas. The transformation of rural land-use into urban one caused an increase of dry-deposition velocities by around 30-50% which alone resulted in a decrease of PM2.5 by 0.1-0.25 $\mu$gm$^{-3}$ in both seasons.



Finally, the impact of biogenic emission modification due to modified land-use and meteorological conditions caused a decrease of summer PM2.5 of about 0.1 µgm$^{-3}$ while the winter effects were negligible. The total impact of urbanization on aerosol components is modelled to be (values indicate winter and summer averages) 0.4 and 0.3 µgm$^{-3}$ for PEC, 0.05 and 0.02 µgm$^{-3}$ for PSO4, 0.1 and 0.08 µgm$^{-3}$ for PNO3, 0.04 and 0.03 µgm$^{-3}$ for PNH4 and 0 and 0.05 µgm$^{-3}$ for SOA. The main contributor of each of these components was the impact of emissions which was usually larger than the total impact due to the fact that UCMF counteracted with a decrease. For each aerosol component the impact of modified DV was a clear decrease of concentration and finally, the modifications of biogenic emissions impacted predominantly SOA causing a summer decrease while a very small secondary effect of secondary inorganic aerosol was modelled too (they increased).

In summary, we showed that when analyzing the impact of urbanization on PM pollution, apart from the impact of emissions and the urban canopy meteorological forcing, one has to consider also the effect of modified land-use and its impact on dry-deposition. These were shown to be important in both seasons. For the effect of modified biogenic emissions, our calculations showed that it acts on PM2.5 predominantly trough SOA modifications which turned to be important only during summer.

## 1 Introduction

In the upcoming years, more the 60% of Earth's population will live in cities (UN, 2018) while urban areas in general represent only a tiny fraction of the habitable land. Moreover, the process of urbanization is predicted to continue by the end-of-the 21st century under all SSPs (socio-economic-pathways) (Gao et al., 2020). It is thus a great desire to quantify the environmental footprints urbanization, or more precisely, the rural-to-urban transformation (RUT) causes.

The RUT acts via two primary pathways: a) with the human activities concentrated in urban areas great amount of emissions are introduced (both green-house-gas and short-lived pollutants) which affect the local but also the regional and global air pollution (Im and Kanakidou, 2012; Markakis et al., 2015; Timothy and Lawrence, 2009; Butler and Lawrence, 2009; Stock et al., 2013; Huszar et al., 2021), b) urban land-surface differs greatly from rural one by introducing artificial object and surfaces with specific geometries (e.g. street canyons, buildings) which affect the surface-air fluxes of energy, momentum and material with strong consequences on meteorological conditions (Oke, 1982; Oke et al., 2017; Karlický et al., 2020) and at the same time, in long-term, they modify the regional climate (Huszar et al., 2014; Karlický et al., 2018). In other words, cities have strong impact on the whole atmospheric environment (Folberth et al., 2015).

Regarding the first pathway, it is clear that urban emissions alone substantially deteriorate the local air-pollution (Thunis et al., 2021). They are composed with a mixture of different gases like oxides of nitrogen (NOx) originating mainly from road transportation (Huszar et al., 2016a, 2021) along with the volatile organic compounds (VOC), carbon monoxide (CO) and sulphur dioxide (SO$_2$) or ammonia (NH$_3$). Further, urban emissions contain also primary aerosol in the form of elemental carbon (PEC), primary organic aerosol (POA) and other primary material like metals (Freney et al., 2014; Allan et al., 2010; Rivellini et al., 2020; Yang et al., 2023). By introducing urban emissions, large quantities of these primary pollutants are added to their background levels. Moreover, they are potentially responsible – as precursors – for the formation of secondary pollutants too. NOx together with VOC (partly supported by CO) leads to formation of ozone (O$_3$) while the ration of NOx-



to-VOC determines the amount of $O_3$ formed or destroyed (Beekmann and Vautard, 2010; Xue et al., 2014). Emissions of gaseous pollutants further lead to formation of secondary aerosol. NOx, $SO_2$ and $NH_3$ are absorbed by water droplets leading to formation of secondary inorganic aerosol (SIA). These include sulphates ($PSO_4$), nitrates ($PNO_3$) and ammonium ($PNH_4$). The main precursor for $PSO_4$ is $SO_2$ which although exhibits decreasing global emissions (Zhong et al., 2020), many urban areas are still marked with significant perturbation of aerosol burden due to this precursor pollutant (Guttikunda et al., 2003;

Yang et al., 20110(@). It has to be further noted that sulfates can be emitted directly too and thus contribute to total particulate matter (PM) pollution (Li et al., 2018). Nitrogen oxides are the main precursors for nitrate aerosol via forming nitric acid ($HNO_3$) which is easily absorbed by water (Seinfeld and Pandis, 1998) and it is well known that urban NOx can significantly contribute to total PM pollution via the formation of $PNO_3$ (e.g. Lin et al., 2010). Ammonia ($NH_3$), while constitutes relatively a small fraction of urban emissions (although there is an indication that transport emits much more ammonia than previously

thought (e.g. Walters et al., 2022)), efficiently helps forming sulfate and nitrate aerosol by reacting to ammonium-sulfates and ammonium-nitrates and is found to be very important in connection with urban emissions (e.g. Behera and Sharma, 2010, and references therein). The thermodynamical equilibrium of the ammonium-sulfate-nitrate-water solution is, in general, rather complicated and highly dependent on ratio of emissions of $SO_2$-NOx-$NH_3$ as well as on the prevailing meteorological conditions (Martin et al., 2004). This gives a potential to high variability of the contribution of different cities to total aerosol

burden.

There is a large number of studies that investigates the perturbation of the atmospheric composition due to the urban emissions: Lawrence et al. (2007), Butler and Lawrence (2009) or Stock et al. (2013) investigated the global impact of emissions from large urban agglomerations. On regional scale, Im et al. (2011a, b); Im and Kanakidou (2012); Finardi et al. (2014); Skyllakou et al. (2014); Markakis et al. (2015); Hodneborg et al. (2011); Huszar et al. (2016a); Hood et al. (2018) looked at

European cities (Paris, London, Istanbul, Athens) but the regional fingerprint of Asian megacity emission has been also of great interest (Guttikunda et al., 2003, 2005; Tie et al., 2013). They all showed that the concentrations of primary pollutants (both gaseous and primary aerosol) are substantially increased locally but also on regional scales. However, on the other hand, secondary pollutant like ozone can respond differently and for these cities often decreases in urban cores are modelled due to high NOx-to-VOC ratio (e.g. Huszar et al., 2016a, 2021). Further, it was found that air pollution in cities is determined mainly

by the local sources, however a considerable part of the total concentration is associated to rural ones (Panagi et al., 2020; Thunis et al., 2021; Huszar et al., 2021).

Besides the direct impact of urban emissions, urbanization influences air chemistry also via the so called "urban canopy meteorological forcing" (UCMF) as introduced by Huszar et al. (2020a). Urban land-surface brings higher temperatures (urban heat island or UHI; (Oke, 1982; Karlický et al., 2020; Sokhi et al., 2022)), drag-induced wind-speed reductions (Huszar et al.,

2018b; Zha et al., 2019) and enhanced vertical turbulent diffusion along with elevated planetary boundary layer height (Ren et al., 2019; Wang et al., 2021). Further it has clear impact on the hydrological cycle by removing the precipitated water via drainage and decreasing thus the humidity over cities (Richard, 2004; Huszar et al., 2018b). UCMF then propagates to modification in transport, deposition and chemical transformation of the emitted pollutants leading to modifications of their concentrations linking the urban meteorological conditions to urban pollution very tightly (Ulpiani, 2021). The impact of





UCMF on air-quality in and around cities (or also rural areas) was a focus of many modelling studies that found that the most important components of UCMF are temperature, wind-speed and turbulence (Struzewska and Kaminski, 2012; Liao et al., 2014; Kim et al., 2015; Jacobson et al., 2015; Zhu et al., 2017; Zhong et al., 2018; Li et al., 2019; Huszar et al., 2018a, 2020a, b; Wei et al., 2018). Due to UCMF, primary gas-phase pollutants and PM are decreased over cities (driven mainly by urban land-surface induced vertical eddy diffusion increase). In case of secondary pollutant the situation is more

difficult as the total impact of UCMF is a combination of the direct impact on the secondary pollutant and the impact on its precursors; e.g. for ozone the resulting effect is an increase (over the surface or at higher levels) (Janssen et al., 2017; Yim et al., 2019; Li et al., 2019; Kim et al., 2021; Kang et al., 2022; Huszar et al., 2022).

Apart from the impact of urban emissions and the impact of UCMF, RUT influences the final air-pollution via two other pathways too. The first is the impact of urbanization induced land-surface change on the dry-deposition of pollutants and

the second is the modification of biogenic emissions due to change (decrease) of vegetation distribution due to urbanization. The land-surface type determines the resistances of that surface (and the canopy layer) which in turn determines the dry deposition velocities (DV) (Zhang et al., 2003; Cherin et al., 2015; Hardacre et al., 2021). It has been shown by many that by urbanization and the consequent reduction of vegetation in urban areas, the deposition velocities are greatly reduced for some gaseous pollutants (e.g. $NO_2$, $O_3$) leading to their increased concentrations (Nowak and Dwyer, 2007; Mcdonald-Buller

et al., 2001; Song et al., 2008; Tao et al., 2015), for others due to higher reactivity on solid surfaces (compared to vegetated ones) the DVs are increased leading to concentration increase (Zhang et al., 2003). For aerosol, DVs are determined mainly by the sedimentation of the particles and by aerodynamic and boundary resistances (Zhang et al., 2001). While sedimentation is determined by particle size and shape, surface resistances are a function of the roughness length and friction velocities (Wesely, 1989) which are enhanced over urbanized land-surface compared to rural ones. This alone would lead to decrease of

PM concentration bearing larger DVs. However, this is modulated also by the modifications of precursor concentrations as a result of the land-surface changes associated with RUT. For example, Huszar et al. (2022) modelled decreases of $NO_2$ and $SO_2$ concentration due to land-surface change (and hence dry deposition modifications) alone which would imply an amplification of the land-surface induced decrease of nitrates and sulphates. Further, it can be assumed that ammonia ($NH_3$) concentrations are also modified by modified dry-deposition velocities which in turn has impacts on the amount of ammonium salts formed.

Regarding the influence of the modified biogenic emissions (BVOC; biogenic volatile organic compounds) as a result of urbanization, one has to realize that the urbanization i) reduces the amount of vegetation (e.g. turning crop land into urban built-up) which alone reduces the emission of biogenic substances (Song et al., 2008). ii) it was detailed above that urban areas exhibit higher-temperature and moreover it seems that cloudiness is somewhat reduced above cities too (w.r.t rural regions) meaning higher solar incident radiation at the surface (Karlický et al., 2020) – both promoting the vegetation metabolism

resulting in higher fluxes of BVOC (Guenther et al., 2006). These two effects (i and ii) counteract but the dominant one is probably the vegetation effect (Li et al., 2019; Huszar et al., 2022), i.e. due to urbanization, BVOC emissions are reduced. As for the effect of such reduction, it is expected that near surface ozone concentration will be decreased as urban areas are usually VOC-limited (Song et al., 2008). In case of PM, this will act via modification of the formation of secondary organic aerosol (SOA). It has been shown by many, that BVOC are an important precursors of SOA and responsible for the formation



of biogenic secondary organic aerosol (BSOA; (Gao et al., 2022)). Couvidat et al. (2013) showed that almost one third of the organic material in the Paris region originates from biogenic VOCs. The great importance of BVOC in urban SOA formation was confirmed also by Sartelet et al. (2012); Zhang et al. (2003); Hu et al. (2017); Nagori et al. (2019); Ma et al. (2023) but of course, anthropogenic precursors remain also very important (Zhang et al., 2015; Guo et a., 2022). While it is clear that most of the BVOC emissions originates from rural and natural land-surfaces (i.e. "non-urban" areas Lin et al., 2016), it is expected

that any change in urban BVOC emissions as a result of urban expansion will have immediate affect on SOA concentrations too, hence, the total PM.

To summarize, urbanization (RUT) greatly modifies the air composition over both the cities themselves but also over surrounding rural areas while this modification is the result of four impacts that add-up to the background air pollution level (i.e. that without urbanization): namely, i) the effect of urban emissions ("DEMIS"), ii) the effect of the urban canopy meteorologi-

140 cal forcing (UCMF) on transport and chemical transformations ("DMET"), iii) the effect of modified dry-deposition velocities as a result of modified (urbanized) land-surface ("DLU_D") and, finally, iv) the effect of modified emissions of BVOC due to modified vegetation-cover and meteorology ("DBVOC"). They together constitute the total RUT impact ("DTOT").

These four impacts had been formulated also in our previous paper Huszar et al. (2022) where we looked, using regional chemistry-transport models coupled to regional climate models, at their effect on gas-phase chemistry, however most of the

145 other studies mentioned above focused either at the total impact of the urbanization or at some of the individual impacts without a detailed analysis of the separate contribution of each of them. The mentioned Huszar et al. (2022) study indeed aimed at the quantification of each individual impact as well as the total impact as one of the first studies of this kind analysing central European domain at moderate horizontal resolution. Here, as a follow-up study, we extend our analysis conducted there to particulate matter and will investigate, how the total PM (as PM2.5, particles of diameter less than 2.5 $\mu m$) as well as their

primary and secondary components respond to these impacts. To fulfill this goal, a background or reference state has to be defined to which these impacts will be gradually included: for our purpose that reference state will be the non-urbanized land-surface without any urban emissions (only rural ones) and without the effect of UCMF. The analysis will focus on present day conditions which includes present day driving climate, emissions and land-use. The four listed impacts will be then gradually added to this reference (i.e. non-urbanized) state in a cascading manner. To consider the uncertainty arising from different

background climate, size and emissions from different cities, we conduct our analysis on a larger selection of 19 cities from central Europe.

As mentioned, our analysis will focus on PM2.5 and its components. Despite notable improvements in European PM pollution, EEA (2022) reports that in 2020, 96 % of the urban population in the European Union was exposed to high levels of PM2.5. This makes the investigation of the components and contributors to urban PM pollution very important. It has to be

also noted that urban air-quality is influenced not only by the local effects. Im and Kanakidou (2012); Huszar et al. (2016a) e.g. showed that emissions from other areas (rural or other, even distant cities) represent a major fraction of urban pollution burdens. Also the UCMF can act on regional scales and the UCMF due to one city can have impact on other ones as shown by Huszar et al. (2014). Here we however will be interested in the local effects only without looking at how the rural emissions contribute to urban air-pollution.



The study is structured in the following way: the Introduction is followed by the presentation of experimental tools (models), their configuration and the data used. Next, the experiments performed are described and the results summarized in the Result section. Finally, these are discussed and conclusions are drawn.

## 2   Methodology

### 2.1   Models used

The study uses the same models, model settings and same input data as Huszar et al. (2022). Here we will therefor summarize only the most relevant information about the model setup stressing the eventual differences.

    The chemistry transport simulations (CTM) were carried out by the CAMx version 7.10 model (Ramboll, 2020) using the Carbon Bond 6 revision 5 (CB6r5) scheme (Cao et al., 2021). Aerosol physics and chemistry were treated with the a static two mode approach and the ISORROPIA thermodynamic equilibrium model (Nenes and Pandis, 1998) was applied for the

175 secondary inorganic aerosol formation. For secondary organic aerosol (SOA), the SOAP equilibrium scheme (Strader et al., 1999) was used. For wet and dry deposition treatment, the Seinfeld and Pandis (1998) and Zhang et al. (2001, 2003) methods were invoked.

    CAMx was driven by the regional climate model (RCM) RegCM version 4.7 (Giorgi et al., 2012) using non-hydrostatic dynamics. PBL physics, cloud and rain microphysics, convection and radiation are treated following Holtslag et al. (1990);

Hong et al. (2004); Tiedtke et al. (1989). The atmosphere-biosphere-surface coupling was treated with the Community Land Model (CLM) version 4.5 (Oleson et al., 2013) land-surface scheme and to account for the urban scale processes, the CLMU module within CLM4.5 was used (Oleson et al., 2008, 2010). CLMU adopts the classical canyon geometry approach, i.e. cities are represented as networks of street-canyons with specified geometry and surface parameters (Oke et al., 2017). RegCM was offline coupled to CAMx using the RegCM2CAMx interface developed by Huszar et al. (2012). The vertical eddy diffusion

coefficients ($K_v$) are diagnosed using the CMAQ approach (Byun, 1999). Because of the offline character of the coupling, no feedback between the RegCM's radiation scheme and chemistry is considered. Based on 10yr long simulations, Huszar et al. (2016b) concluded that the radiative effects of urban pollutant emissions and secondarily formed pollutants are minor which justified the offline coupling.

### 2.2   Model setup and data

Model simulations with RegCM and CAMx were conducted over 9 km x 9 km resolution domain covering "larger" central Europe (from France to Ukraine and from northern Italy to Denmark) with $189 \times 165$ gridboxes, centered over the Czech capital, Prague (50.075° N, 14.44° E; Lambert Conic Conformal projection). In vertical, the model grid spawns 40 layers in RegCM up to 5 hPa, while CAMx uses the lowermost 18 layers up to about 12 km. The simulated years are DEC 2014 to DEC 2016 with the first month as spin-up. Fenech et al. (2018) showed that the difference between the coarse and fine resolution

PM2.5 concentrations is rather small and our resolution is comparable to the resolution of the emissions (see below). The





requirement of Tie et al. (2010) that the size of the city to resolution should be 6:1 means that preferably, a resolution 6 km or finer should be used, however we rely on the fact that land-use is represented as fractional land-use so even the smallest cities are resolved within the surface model in our RCM. Markakis et al. (2015) showed that the modelled PM2.5 concentrations for Paris are more sensitive to the emissions inventory resolution than on the resolution at which the meteorology is resolved in the driving RCM.

RegCM simulations are forced with the ERA-Interim reanalysis (Simmons et al., 2010). Chemical boundary conditions are taken from the CAM-chem global climate-chemistry model (Buchholz et al., 2019; Emmons et al., 2020). Land-use for RegCM and for the CAMx dry-deposition calculations are based on the 100 m resolution CORINE CLC 2012 landcover data (https://land.copernicus.eu/pan-european/corine-land-cover, last access May 16, 2023) except a small area over Belarus, where CORINE is not available, so the United States Geological Survey (USGS) data is used.

Anthropogenic emissions are taken from the European CAMS (Copernicus Atmosphere Monitoring Service) version CAMS-REG-APv1.1 inventory (Regional Atmospheric Pollutants; (Granier et al., 2019)) for year 2015 combined with the Czech national emission data, the Register of Emissions and Air Pollution Sources (REZZO) dataset issued by the Czech Hydrometeorological Institute (www.chmi.cz, last access May 16, 2023) and the ATEM Traffic Emissions dataset provided by ATEM (Ateliér ekologických modelů – Studio of ecological models; www.atem.cz, last access May 16, 2023). These annual sector based emission totals are decomposed to hourly speciated emissions fluxes using the Flexible Universal Processor for Modeling Emissions (FUME) emission model (http://fume-ep.org/; Benešová et al., 2018) using speciation and time-disaggregation factors of Passant (2002) and van der Gon et al. (2011). To account for the SOA formation from intermediate VOC (IVOC) which are normally not included in emission inventories, we proceeded following Ciarelli et al. (2017); Giani et al. (2019) to calculate IVOC based on the known NMVOC and POA (primary organic aerosol) emissions from gasoline and diesel vehicles, and emissions from biomass burning. This emission adjustment was not implemented in Huszar et al. (2022) but there we looked at gas-phase chemistry impacts only.

Biogenic emissions are computed offline using MEGANv2.1 (Model of Emissions of Gases and Aerosols from Nature version 2.1) with the algorithm described by Guenther et al. (2012) driven by RegCM meteorological fields (short-wave radiation, temperature, soil moisture, humidity). The necessary input for MEGAN were not part of the CORINE land-use information and were compiled based on Sindelarova et al. (2014, 2022). These include leaf-area index data (weekly data), plant functional types and emission potentials of different plant types. Besides BVOC, MEGAN also calculates the fluxes of soil-biogenic NO (nitrogen monoxide) emissions from bacterial activity (Yienger and Levy, 1995). As these emissions are a function of LAI and meteorological conditions, a fraction of the "DBVOC" impact will be composed of soil-NOx emissions modifications. Not presented here, in our experiments the soil-NOx emissions were about two orders of magnitude smaller compared to the BVOC emissions and their effect is expected to be much smaller including the effect of their urbanization induced modulations. BVOC emission fluxes are strongly temperature dependent, i.e. higher temperatures result in enhanced emissions. In this regard, it is expected that urbanization induced temperature increase will lead to higher BVOC fluxes.

For the purpose of the study, we had to isolate the emissions originating from selected urban areas from rural emissions. To achieve this, we used the masking capability of the emissions model used (FUME) while we used the administrative boundaries



of the chosen cities based on the GADM public database (https://gadm.org, last visit May 16 2023) which provide geographic shape-files of these boundaries. Cities selected in the analysis are Berlin, Brussels, Budapest, Cluj-Napoca, Cologne, Frankfurt, Hamburg, Krakow, Lodz, Lyon, Milan, Munich, Prague, Torino, Vienna, Warsaw, Wroclaw, Zagreb, Zurich. The choice considered the same criteria as in Huszar et al. (2021, 2022): the diameter of the city larger than 9 km (the gridcell size in our model), minimal orographic variability to reduce orographic effects (see e.g. Ganbat et al., 2015), sufficiently large distance between cities eliminating mutual influences and, finally, no coastal cities to eliminate the effect of asymmetric land-use, like e.g. the sea-breeze effect (e.g. Ribeiro et al., 2018). Although strict emission control policies, these cities are still often burdened with high air pollution due to PM (Khomenko et al., 2021; Sokhi et al., 2022; Balamurugan et al., 2022; Putaud et al., 2023).

## 2.3 Model simulations

In a similar fashion that in (Huszar et al., 2022), we decomposed the total impact of urbanization (RUT) into the individual impacts or contributors (i.e. "DEMIS", "DMET", "DLU_D", "DBOC") listed in the introduction. This required to carry out a series of model simulations with each contributor added gradually one-by-one to the reference simulation arriving to the full urbanization case.

First, we carried out a pair of model simulations with RegCM: "Urban" and "Nourban" with former accounting for urban land-surface treated with the RegCM's urban canopy module while in the later, land-use was replaced by "crops" as the most common rural land-use type in the region analyzed. We performed 5 simulations with CAMx that differ in the in-/exclusion of urbanized land-surface, the UCMF (acting on both atmospheric chemistry in general and on biogenic emissions) and urban emissions. These simulations are summarized in Tab. 1. The first simulation called "ENNNN" represents the hypothetical reference (background) state without urban emissions and with the urban land-surface replaced by rural land-surface in RegCM and CAMx as well as in the BVOC calculations (with MEGAN). In the 2nd experiment, "ENYNN", the urban emissions are turned on. In the 3rd experiment, "ENYUN", the urban land-use was "switched-on" for the dry-deposition scheme in CAMx. In the 4th experiment, "ENYUU", urban land-use and the UCMF (i.e. "Urban" meteorology) is accounted for the biogenic emissions model and finally, in the 5th experiment, "EUYUU", all the urbanization-related effects are considered, representing the most realistic full case.

In the first experiment where urban emissions are disregarded, we removed urban emissions only for the 19 cities selected. For the effect of rural-urban land-use transformation on meteorological conditions, dry-deposition and biogenic emissions, we replaced the urban land by rural one over the entire domain (i.e. not only for the cities selected). It is clear that this has effect on the background level of air pollutants and not only on local urban levels, but the effect is probably much smaller than local effects as 1) emissions from these areas were still considered, 2) the urban meteorological effects from these (minor) urban areas have rather small influence or air pollutants as the UCMF over them is also small (see e.g. Huszar et al. (2014)).





Similar as in Huszar et al. (2022), we can express mathematically the concentrations $c_i$ of a pollutant $i$ in a selected city with respect to RUT in a following way:

$$c_i = c_{i,rural} + \Delta c_{i,RUT}, \tag{1}$$

where $c_{i,rural}$ is the reference (background) concentration before RUT and $\Delta c_{i,RUT}$ is the total impact of urbanization.

In this study, we are concerned about the contributors to $\Delta c_{i,RUT}$ (regardless of their sign), i.e.:

$$\Delta c_{i,RUT} = \Delta c_{i,EMIS} + \Delta c_{i,MET} + \Delta c_{i,LU_D} + \Delta c_{i,BVOC}, \tag{2}$$

where $\Delta c_{i,EMIS}$, $\Delta c_{i,MET}$, $\Delta c_{i,LU_D}$ and $\Delta c_{i,BVOC}$ are the impacts of urban emissions, the impact of the urban canopy meteorological forcing, the impact of modified land-use on dry-deposition and the impact of modifications of BVOC emissions,
denoted above as "DEMIS", "DMET", "DLU_D" and "DBVOC".

These impacts will be calculated from the experiments listed in Tab. 1 as indicated below (the experiment number is shown in parenthesis):

$$\begin{aligned}
\Delta c_{i,RUT} &= EUYUU(5) - ENNNN(1) \\
\Delta c_{i,EMIS} &= ENYNN(2) - ENNNN(1) \\
\Delta c_{i,MET} &= EUYUU(5) - ENYUU(4) \\
\Delta c_{i,LU_D} &= ENYUN(3) - ENYNN(2) \\
\Delta c_{i,BVOC} &= ENYUU(4) - ENYUN(3)
\end{aligned} \tag{3}$$

It has to be realized that, in fact, the contributors above act simultaneously and feedback are present between them so their
impacts are not additive. The way how we calculated them however allow us to consider them to be additive meaning that their sum is the total impact of urbanization. This is also a consequence of Eq. 3.

Our analysis will focus on near surface PM2.5 concentrations as well as their secondary components, i.e. secondary inorganic aerosol (SIA) represented by sulphates ($PSO_4$), nitrates ($PNO_3$) and ammomium ($PNH_3$) and secondary organic aerosol. Moreover, we will also focus on primary elemental carbon (PEC) which is an important fraction of urban emission loads. As
the emissions of primary organic aerosol have very similar magnitude in our emission data compared to PEC and it has the same deposition velocity (which is determined only by size), we will not explicitly analyze POA concentrations as we assume that the impacts on POA will be very similar to PEC.

## 3 Results

### 3.1 Validation

Here we compare the modelled concentrations of PM2.5 and their components (PSO4, PNO3, PNH4, SOA and PEC). The measured data for PM2.5 are taken from the AirBase European air quality data (http://www.eea.europa.eu/data-and-maps/





data/aqereporting-1, last access 16 May 2023) while for PM components data are taken from EBAS database (https://ebas-data.nilu.no/, last access May 16 2023) from EMEP background sites. AirBase data are taken from all available rural and urban background stations in order to distinguish between model performances above both type of stations.

Fig. 1 shows the average annual cycle of monthly means for urban and rural stations including the corresponding model values. Over urban stations, CAMx exhibit a strong underestimation around 5-10 $\mu gm^{-3}$ being stronger in winter. CAMx performs slightly better over rural stations with smaller negative bias. In both cases, the annual cycle is reasonably captured. We also compared the analysed components (sulphates, nitrates, ammonium and elemental carbon) with measurements. In this case the EMEP background station data acquired from the EBAS (https://ebas.nilu.no/data-access/, last access 5 May 2023)

was used. Fig. 2 shows that the model underestimates PSO4 by about 0.5-1 $\mu gm^{-3}$, especially during summer when the model predicts minimum values while in measurements, the values show more or less uniform distribution during the year. PNO3 is overestimated during winter by about 2 $\mu gm^{-3}$ and the model well matches the summer values. In case of PNH4, an 0.5 $\mu gm^{-3}$ overestimation of winter values and a similar underestimation of summer values is encountered. Thus, for both PNO3 and PNH4, the amplitude of the annual cycle is overestimated. For PEC, the match is very satisfactory with an uniform model

bias of -0.25 $\mu gm^{-3}$ throughout the year meaning that the annual cycle is very well captured by the model.

## 3.2   The impact of individual contributors to RUT

Here we present the total impact of RUT as well as its individual contributors on PM2.5 concentrations as 2015-2016 DJF and JJA averages, averaged across the selected cities. Values from model gridboxes that cover the city centres are selected. The results are shown as boxplots in Fig. 3; they show the 1st and 3rd quartiles and the median values along with the minimum and

maximum value across all the cities.

        As expected, the highest impact is attributed to the effect of emission only causing an increase of urban concentrations by around 1.5 to 3.4 $\mu gm^{-3}$ in DJF and by about 1.2 to 2 $\mu gm^{-3}$ in JJA. The effect of UCMF on concentration is usually a decrease up to -1 and -0.4 $\mu gm^{-3}$ in winter and summer, respectively. The impact of minor contributors associated with modified land-use and deposition velocities and modified BVOC emissions are a decrease for "DLU_C" for both seasons by

310     -0.2 to -0.3 in DJF and -0.08 to -0.15 in JJA, while for the effect of BVOC modification is very small, around -0.05 in summer and almost zero in DJF. The total impact of urbanization on PM2.5 is 1.2 to 3 $\mu gm^{-3}$ increase in DJF and a smaller increase, from 1 to 1.6 $\mu gm^{-3}$ in JJA, while of course, the impact of emissions dominates.

        To examine the contribution of the most important aerosol components to these changes, we plotted a similar figure as above but individually for PSO4, PNO3, PNH4, PEC and SOA presented in Fig. 4. Sulphates respond to urban emissions alone by

an increase by about 0.05-0.15 $\mu gm^{-3}$ and up to 0.05 in DJF and JJA, respectively. The urban canopy meteorological forcing results in a decrease of sulphates in DJF by up to -0.05 $\mu gm^{-3}$ while increases are modelled for JJA up to 0.025 $\mu gm^{-3}$. The impact of dry-deposition change on PSO4 is a decrease: up to -0.025 in DJF and -0.01 in JJA. The BVOC effect on PSO4 is a slight increase in JJA and virtually zero in DJF. The total impact of RUT on PSO4 is an increase up to 0.15 and 0.05 in DJF and JJA, respectively.





In case of nitrates, the urban emissions alone increase urban concentrations by about 0.1 to 0.2 µgm$^{-3}$ in both seasons (somewhat more in DJF). The effect of UCMF is an increase in DJF by about 0.05 µgm$^{-3}$ while in JJA, decreases dominate up to -0.15 µgm$^{-3}$. The impact of modified dry-deposition is a decrease by about -0.08 to -0.11 in DJF and -0.03 to -0.06 in JJA. The impact of modified BVOC emissions are negligible and the total impact of RUT on PNO3 is an increase up to about 0.16 and 0.12 in DJF and JJA, respectively.

For ammonium, urban emissions cause increase by 0.04 to 0.09 µgm$^{-3}$ and by 0.04 to 0.05 µgm$^{-3}$ in DJF and JJA, respectively. The sign of the UCMF impact can be positive and negative in both seasons with values between -0.01 to 0.025 µgm$^{-3}$ in DJF and between -0.03 to 0.02 µgm$^{-3}$ in JJA. The impact of dry-deposition modification is negative in both seasons with -0.03 to -0.04 µgm$^{-3}$ and -0.01 to -0.02 µgm$^{-3}$ decrease in winter and summer, respectively. The impact of modified BVOC emissions are again negligible. Finally, the total impact of RUT on PNH4 is an increase up to about 0.04 and 0.03 in DJF and

JJA, respectively.

   The impact on elemental carbon which is a primary component chemically inert in CAMx (with no chemical decay or reactions) is as follows: urban emissions cause increase of PEC by around 0.2 to 0.5 µgm$^{-3}$ in DJF and around 0.2 to 0.4 µgm$^{-3}$ in JJA. The UCMF causes a slight decrease of PEC by around -0.05 to -0.1 µgm$^{-3}$ in both seasons. The increased deposition velocities caused decreased concentrations of PEC by about -0.01 to -0.015 µgm$^{-3}$ in DJF and around -0.005

µgm$^{-3}$ in JJA. Being an inert PM component in CAMx, no impacts of BVOC modifications on PEC are modelled. The total impact of RUT on PEC is again dominated by urban emissions and reached 0.6 µgm$^{-3}$ and 0.3 µgm$^{-3}$ in DJF and JJA, respectively.

   Finally, the impact secondary organic aerosol has considerable values only during JJA when dominantly the oxidation of primary VOC to semivolatile precursors of SOA take place. The impact of urban emissions is an increase of SOA by up to 0.05

to 0.1 µgm$^{-3}$ while urban meteorological changes cause SOA modifications usually between zero and 0.05 µgm$^{-3}$. Due to land-use modifications and associated deposition velocity increases, SOA responds with a decrease up to -0.02 µgm$^{-3}$ while due to urbanization induced BVOC modifications, SOA decreases by around -0.04 to -0.06 µgm$^{-3}$. The total impact of RUT on SOA is an increase by 0.07 µgm$^{-3}$ in JJA and a very tiny increase by up to 0.01 µgm$^{-3}$ in DJF.

### 3.3 The spatial distribution of the impacts

To obtain a spatially resolved information about the impact of individual contributors to RUT, we plot here their 2-D distribution as DJF and JJA averages. We start with presenting the distribution of absolute concentrations for comparison of the changes with absolute values in order to resolve the relative magnitude of these contributors.

   Fig. 5 shows the absolute near surface concentrations of DJF and JJA PM2.5 and its analyzed components. The PM2.5 concentrations reach 30-40 µgm$^{-3}$ in winter while rural areas are often over 8 µgm$^{-3}$. In summer, concentrations are, as

expected, smaller, reaching 10 µgm$^{-3}$ being above 4 µgm$^{-3}$ over rural areas. In DJF, the highest contribution is made by nitrates reaching 20 µgm$^{-3}$ over northern Italy and being about 4-6 µgm$^{-3}$ over central Europe. The concentrations of sulphates are large especially over Poland, reaching 2-3 µgm$^{-3}$ while ammonium is largest over northern Italy reaching 4-6 µgm$^{-3}$ while other areas exhibit concentrations around 1-2 µgm$^{-3}$. Elemental carbon contributes to total PM2.5 by values around 2-4





μgm$^{-3}$ over northern Italy while the contribution is clearly limited to urban areas over other regions within the domain (e.g. 1-2

μgm$^{-3}$ over urban areas in eastern Europe). The SOA concentrations in JJA are usually between 0.2-1 μgm$^{-3}$ reaching maxima again over Italy (2-3 μgm$^{-3}$). In summer, the secondary inorganic aerosol concentrations are somewhat smaller, especially for ammonium (less than 0.6 μgm$^{-3}$) while nitrates are largest over western Europe reaching 2-3 μgm$^{-3}$ and sulphates largest over southern Europe, reaching also around 2-3 μgm$^{-3}$. The PEC concentrations in JJA are small, usually around 0.1 to 0.4 μgm$^{-3}$. SOA is larger in summer than in winter reaching concentration up to 2 μgm$^{-3}$ and being usually around 0.4-1.5 μgm$^{-3}$.

### 360 3.3.1 The impact of urban emissions (DEMIS)

In Fig. 6 the DJF and JJA average spatial impact of urban emissions ("DEMIS") on the near-surface concentrations of PM2.5 and its five analysed components are presented. Urban emission impact is in general higher in winter expect for SOA. In DJF, PM2.5 are increased over urban areas by up to 4 μgm$^{-3}$ and the contribution to rural concentrations can also reach 0.5 μgm$^{-3}$. In JJA, urban emission contribute to total PM2.5 by about 1-3 μgm$^{-3}$ over cities while the rural contribution is small, reaching

0.02 μgm$^{-3}$. The impact of urban emissions on PEC reaches 1 and 0.5 μgm$^{-3}$ in city centres in DJF and JJA, respectively, while the impact over rural areas is less than 0.02 μgm$^{-3}$. For sulphates, urban emission impact urban concentrations up to 0.5 μgm$^{-3}$ in winter while the summer impact is smaller reaching 0.1 μgm$^{-3}$. The impact over rural areas is again less than 0.02 μgm$^{-3}$ in both seasons. Much larger impact is modelled for nitrates reaching 1 μgm$^{-3}$ over Italy and exceeding 0.2 over most of central Europe in winter. In summer, the impact on PNO3 is smaller reaching 0.5 μgm$^{-3}$ over urban- and 0.2 μgm$^{-3}$

over rural areas. The impact of urban emissions on ammonium reaches 0.5 μgm$^{-3}$ and is usually around 0.05 μgm$^{-3}$ in DJF while JJA concentrations are smaller reaching 0.05 but usually less than 0.02 μgm$^{-3}$. Finally, the impact on SOA is negligible in winter reaching 0.02 over city centres while in summer, it can reach 0.2 μgm$^{-3}$ and the contributions over rural areas can exceed 0.02 μgm$^{-3}$.

### 3.3.2 The impact of modified meteorological conditions (DMET)

In Fig. 7 the DJF and JJA average spatial impact of the urban canopy meteorological forcing ("DMET") is presented. For PM2.5, it is characterized by decreases located above urban areas reaching -2 μgm$^{-3}$ in both seasons. Elsewhere, i.e. above rural areas, PM2.5 increases by up to 1-1.5 μgm$^{-3}$. In case of PEC, the decrease over urban areas are evident and reaches -0.2 μgm$^{-3}$, especially during DJF, while some minor increases are modelled over rural land reaching 0.05 μgm$^{-3}$. The impact for secondary aerosol components is more complicated as apart from the direct impact, UCMF impacts also their precursors.

Sulphates decrease above urban areas by about 0.2 to 0.5 μgm$^{-3}$ in DJF and by 0.1 in JJA. Large rural regions show, on the other hand, increases of PSO4 by up to 0.1-0.2 μgm$^{-3}$, mainly in winter. In case of nitrates, some urban areas exhibit decreases in DJF (e.g. Berlin, the Ruhr area) but also large increases are modelled over rural areas and even urban ones, especially over northern Italy along the Po river reaching 0.5 μgm$^{-3}$. In summer, the decrease over urban areas is seen for most of the cities, however, over other areas, a strong increase of PNO3 is modelled reaching 0.5 μgm$^{-3}$. The UCMF's impact on PNH4 is

somewhat smaller and is, again, characterized by decreases above cities up to -0.2 μgm$^{-3}$ in both seasons, while increases are modelled over rural areas and also some urban ones reaching 0.3 μgm$^{-3}$ in winter and 0.2 μgm$^{-3}$ in summer. In case of



SOA, some urban areas over western and southern Europe exhibit decreases in JJA up to -0.05 $\mu$gm$^{-3}$ but increases dominate reaching 0.2 $\mu$gm$^{-3}$. In DJF, the impact is very small with some minor increase over rural areas up to 0.05 $\mu$gm$^{-3}$.

### 3.3.3   The impact of dry-deposition modifications (DLUC_D)

Fig. 8 depicts the DJF and JJA average spatial impact of the modified dry-deposition velocities due to urban land-use. In both seasons, a clear decrease of concentrations is modelled indicating that dry deposition velocities increased, as it was expected. The total PM2.5 concentrations decreased by up to 1.5 $\mu$gm$^{-3}$ in winter over cities while large rural areas exhibit a decrease up to -0.5 $\mu$gm$^{-3}$. In summer, the decreases have smaller magnitude, reaching -0.5 $\mu$gm$^{-3}$ over urban areas while over rural ones they reach -0.2 $\mu$gm$^{-3}$. For PEC, decreases are limited mostly to urban areas reaching -0.05 $\mu$gm$^{-3}$ in DJF and -0.01 $\mu$gm$^{-3}$

in JJA. Larger impact are modelled for secondary aerosol, probably due to the fact that their precursors are also impacted. Sulphates decreased in winter by about 0.05 $\mu$gm$^{-3}$ and by about 0.02 $\mu$gm$^{-3}$ in summer, mainly over urban areas. Among SIA, the largest impacts are modelled for nitrates, exceeding -0.1 $\mu$gm$^{-3}$ decrease in DJF over northern Italy but being large over rural areas too (about -0.05 $\mu$gm$^{-3}$). In summer, PNO3 decreases by around 0.02 to 0.05 $\mu$gm$^{-3}$, mainly over cities. In case of PNH4 the decreases are, again, largest above cities reaching -0.05 $\mu$gm$^{-3}$ in both seasons (slightly stronger decrease

in DJF). Over rural areas, the decrease is about -0.02 $\mu$gm$^{-3}$ and -0.01 $\mu$gm$^{-3}$ in DJF and JJA, respectively. SOA decreased due to modified dry-deposition velocities by around 0.02 $\mu$gm$^{-3}$ above cities in both season, while over rural areas, it reaches about -0.01 $\mu$gm$^{-3}$, slightly higher in JJA.

The decrease above are the result of increased deposition velocities, which are depicted in Fig. 9 for PEC for winter and summer. As the DVs in the model used (CAMx) are a function only of the size, all aerosol within the 0-2.5 micron size range

(where all the secondary aerosols belong) have the same DV values, we present here only the modification of DV for only this component (for others the figure would be the same). DVs increased clearly above urban areas while the increase reaches 0.1 cms$^{-1}$ in DJF for some cities. For JJA, the increases are slightly smaller, usually between 0.02 and 0.1 cms$^{-1}$.

### 3.3.4   The impact of biogenic emissions (DBVOC)

Fig. 10 presents the impact of modified biogenic emissions due to RUT on PM concentrations. It is clear that BVOC emissions

are important mainly during the warm seasons and that is why the impacts during DJF are much smaller than during JJA. Moreover, during summer BVOC can more readily oxidize to semi-volatile hydrocarbons forming SOA so the impact on PM2.5 act predominantly via impacting secondary organics concentrations. However due to feedback on the overall gas-phase chemistry and thus SIA precursors, SIA are also slightly modified. In winter the impact on PM2.5 is considerable only above norther Italy reaching -0.05 $\mu$gm$^{-3}$ while mainly SOA and nitrates contribute to these PM modifications. In case of PNO3,

they decrease above the same region by around 0.01-0.02 $\mu$gm$^{-3}$ while SOA decreased by a similar magnitude (and slightly increased over other areas). Sulphates responded to BVOC changes by a slight decrease up to -0.01 $\mu$gm$^{-3}$. In JJA, the impacts are in general much larger (as expected) and are mainly determined by the decreased SOA but also modulated by increases in SIA. The PM2.5 JJA decrease reaches -0.1 $\mu$gm$^{-3}$ (again mainly over northern Italy) but is between -0.02 to -0.05 $\mu$gm$^{-3}$ over large areas all over the domain. Regarding SIA, all of them increased: by up to 0.02 $\mu$gm$^{-3}$ for PNO3 and up to 0.01





μgm$^{-3}$ in case of PSO4 and PNH4. For SOA there is a clear decrease during JJA up to -0.1 μgm$^{-3}$ over northern Italy and being between -0.02 and -0.05 μgm$^{-3}$ over large regions across the domain. As PEC is not affected by either gas-phase and aerosol chemistry, no modifications due to biogenic emission changes are modelled.

## 3.4 The diurnal variation of the impacts

Human activities change during the day causing a typical diurnal cycle of urban emissions. Moreover, the urban canopy
meteorological forcing has also a distinct diurnal pattern, e.g. the modification of temperature is strongest during night, the impacts on wind and turbulence are the strongest during day-time and so on (Huszar et al., 2018a). It is thus expected that the individual contributors to the total impact of RUT analyzed here will have also a distinct diurnal cycle.

In Fig. 11 and 12 we present the diurnal cycles for the four contributors' impact on PM2.5 and its components during winter and summer averaged over all urban centres (we took the model gridbox covering the city centre, in a similar way
as in Fig 3). In case of PM2.5 in winter, "DEMIS" causes a typical diurnal variation resembling the diurnal cycle of urban emissions (varying between 1.5 and 4 μgm$^{-3}$), this is also seen for PEC, when maxima occur during morning and evening rush hours. A similar diurnal pattern is seen also for sulphates varying between 0.1 and 0.2 μgm$^{-3}$. For other secondary aerosol components the diurnal cycles are characterized by only one maximum: for nitrates, the maximum occurs during noon time reaching 0.25 μgm$^{-3}$, while for ammonium, the maximum emission impact is reached during morning reaching 0.1 μgm$^{-3}$.
SOA are increased due to emission at most during early afternoon by up to 0.015 μgm$^{-3}$. In case of the impact of UCMF ("DMET"), it is usually negative for PM2.5 being lowest during afternoon reaching around -1 μgm$^{-3}$. For PEC and PSO4, the maximum decrease is about -0.2 and -0.1 μgm$^{-3}$, respectively. For PNO3 and PNH4 and SOA, it is again negative during afternoon hours reaching -0.2, -0.07 and -0.02 μgm$^{-3}$, respectively. The impact of increased deposition velocities is negative in all cases and throughout the whole day in winter. However, the diurnal patterns indicate that the maximum decrease is modelled
for early afternoon hours, reaching -0.3 μgm$^{-3}$ for PM2.5. For PEC, PSO4, PNO3 and PNH4, it reaches -0.02, -0.02, -0.01 and -0.04 μgm$^{-3}$, respectively. For SOA, the maximum decrease reaches -0.016 μgm$^{-3}$. As expected, the impact of modified BVOC emissions is almost negligible with weak maximum decrease during afternoon and evening hours for SOA (around -0.002 μgm$^{-3}$).

The JJA diurnal cycles of the impacts are similar to DJF in case of "DEMIS" with two maxima for PM2.5, PEC and
445 PSO4, while a single maximum due to emissions is modelled for the other components. For the "DMET", again an early evening decrease is modelled reaching -0.5 and -0.1 μgm$^{-3}$ for PM2.5 and PEC. The impact on PSO4 is very small reaching -0.01 μgm$^{-3}$, while for PNO3, PNH4 and SOA, the maximum decrease is about -0.15, -0.04 and -0.05 μgm$^{-3}$, respectively. The summer "DLU_D" impact on PM2.5 and its components has a distinct cycle compared to DJF with usually a morning maximum decrease. This reaches -0.16 μgm$^{-3}$ for PM2.5. It is further very small for PEC. For PSO4, PNO3, PNH4 and SOA
reaching -0.02, -0.15, -0.03 and -0.025 μgm$^{-3}$, respectively. In contrary to DJF, the impact of biogenic emissions changes due to urbanization show a clear diurnal cycle for all PM components except PEC, which does not interact with gas-phase species. For PM2.5, concentrations decreased and this decrease is at its maximum during evening hours reaching -0.08 μgm$^{-3}$. For PSO4, increases are modelled reaching their maximum during noon time (0.005 μgm$^{-3}$), while for PNO3 and PNH4, increase



is modelled too, but during afternoon there is a slight decrease of nitrate concentrations. In case of SOA, a strong decrease is

455 modelled during evening hours reaching -0.075 $\mu gm^{-3}$ which clearly determines the overall cycle for total PM2.5. For other

hours, the decrease of SOA in JJA is around -0.05 $\mu gm^{-3}$.

The presented diurnal variations are in close relation to diurnal pattern of emissions, both anthropogenic (in case of "DEMIS")

and biogenic (in case of "DBVOC") and also to the diurnal cycle of UCMF (in case of "DMET"). While for emissions, these

are more or less well known (maximum during daytime for BVOC and two daytime maxima in case of anthropogenic emis-

460 sions), for UCMF the underlying causes are hidden in the diurnal pattern of individual components in UCMF (temperature,

wind, planetary boundary layer height, turbulence etc.) and these have been modelled and well described in our previous stud-

ies (Huszar et al., 2018a, b, 2020a). However, the diurnal patter of deposition velocities for PM and their urbanization-induced

modifications have not yet been evaluated. Therefor here, to accompany the diurnal variation of concentrations, we plot the

diurnal variation of the changes of DV due to urban land-surface as well as the absolute values (corresponding to non-urbanized

land-surface). We chose PEC as a representative PM component (note, that DV are a function of PM size only and we consider

only fine aerosol in this study). The results are depicted in Fig. 13. The absolute DVs range between 0.8 and 1.2 $mms^{-1}$ and

0.8 and 1.5 $mms^{-1}$ in DJF and JJA, respectively. The maximum values are reached during noontime. The changes due to the

introduction of urban land-surface follow a very similar pattern with the highest impact during noon-time reaching 0.8 and 0.5

$mms^{-1}$ in DJF and JJA, respectively and being about 0.5-0.6 and 0.3 $mms^{-1}$ during night-time for both seasons.

**4    Discussion and conclusions**

In this study, an analysis of the different contributors to the overall impact of urbanization (called rural-urban-transformation;

RUT) on particulate matter concentrations over central Europe was presented. It focused on the four most important contribu-

tors to RUT: the impact of urban emissions only ("DEMIS"), the impact of the UCMF (urban canopy meteorological forcing)

on PM transport and chemistry ("DMET"), the impact of modified dry-deposition velocities due to the urbanized land-cover

("DLU_D") and the impact of modified biogenic emissions due to modified land-cover (and associated vegetation change) and

modified meteorological conditions ("DBVOC"). They were quantified by performing a set of model simulations were each of

the contributors were added one-by-one starting with the reference state corresponding to non-urbanized land-surface with no

urban emissions.

The model biases identified for PM2.5 show that some of the PM2.5 components are strongly underestimated in CAMx.

Very similar underestimation was encountered previously by Huszar et al. (2021) for selected European cities, using the same

resolution and same emission data (they used however and older version of the models). Ďoubalová et al. (2020) too using

CAMx reported for a central European a comparable negative bias. From the analysed aerosol components, sulphates were

underestimated but this along with the model performance for nitrates and ammonia does not explain the strong negative bias

for PM2.5. Most probably, the organic aerosol fraction was strongly underestimated in our model, which is a general problem

in CTMs and has been encountered previously by many authors (Jiang et al., 2019; Ciarelli et al., 2017), while, at the same

time, it is considered as often the largest fraction of fine particulates in urban environments (Allan et al., 2010; Lanz et al.,



2010). It also probable that the bias is partly caused by the fact the the local wind blow dust sources are not considered in our emissions model while this dust source can significanly contribute to overall PM2.5 over central Europe as shown by Liaskoni et al. (2023). The modelled negative bias for sulphates is similar to Bartik el al. (2021) who applied CAMx over a similar

European domain using slightly newer emission data. Sulphates were moreover underestimated over Europe by the majority of the models used in the AQMEII phase 2 model inter-comparison (Im et al., 2015) and can be connected to overestimation of other secondary inorganic components, i.e. ammonium preferably neutralizes nitrates instead of sulphates leading to less ammonium sulphate formation (Im et al., 2015).

   Due to the overall underestimated PM2.5 concentrations, it is expected that the impacts presented here for PM2.5 are un-
derestimated too. This is true especially for the impact of emissions but as the effect of UCMF and deposition velocity change is also proportional to the concentrations they act on, the impacts presented below are underestimated most probably in these cases too.

   The total impact of urbanization on PM2.5 over central European cities was calculated to be 1-1.5 $\mu\mathrm{gm}^{-3}$ in average in summer while in winter, urbanization increased air pollution even more, by around 2-3 $\mu\mathrm{gm}^{-3}$. When comparing these result
to other similar studies one has to remember that this includes not only the impact of urban emissions but also the counteracting effect of UCMF and increased dry-deposition, and, in a minor way also the impact of biogenic emissions changes. Indeed, urban emission alone increased PM2.5 concentration by 1.2-2 $\mu\mathrm{gm}^{-3}$ and 1.5-3.5 $\mu\mathrm{gm}^{-3}$ in winter and summer. These number are close to what was modelled in Huszar et al. (2016a) who looked only at the effect of emissions. The reason we got slightly smaller numbers is that they used 2005 emissions which were higher compared to our 2015 emissions used in this study. Our
PM2.5 increases due to urban emissions only are also much lower than in Im and Kanakidou (2012) but they modelled Istanbul and Athens which are large megacities, much larger than the average of our central European selection. Previously, Skyllakou et al. (2014) showed for Paris that the contribution of local sources to PM is smaller, around 1 $\mu\mathrm{gm}^{-3}$ however they used much coarser resolution and could not thus capture the city core contributions. Indeed, the contributions of urban emissions to urban air-pollution over the urban core is much larger if higher resolutions are applied, as seen in Huszar et al. (2021) as
the largest contributions occur over city centres (Thunis et al., 2021). Finardi et al. (2014) made estimates on the impact of Po valley, a highly urbanized region in Northern Italy on PM2.5 concentration and they found that local emissions contribute to local concentration by up to 10-20 $\mu\mathrm{gm}^{-3}$ which is much larger contribution that our 4 $\mu\mathrm{gm}^{-3}$ simulated for Milan (as city belonging to this region) however, they simulated the contribution from a much larger, regional emissions source and not only one city.

As already seen, the total urban impact is lower if other contributors besides the urban emissions are considered. From these, the impact of UCMF showed a clear decrease of PM2.5 concentrations by around 0.5 $\mu\mathrm{gm}^{-3}$ counteracting the increase due to urban emissions. Indeed, many showed that the most important component of UCMF is the enhanced vertical eddy-transport which removes pollutants from the surface layer (where they are emitted) causing decreased concentrations (Kim et al., 2015; Huszar et al., 2018b, 2020a; Zhu et al., 2017; Wei et al., 2018). Moreover, our spatial results showed that PM2.5 increases due
to UCMF over rural areas, which was also seen in (Huszar et al., 2018b) and is probably the result of the fact that the removed PM by increased turbulence is deposited to lower model levels over other regions further from the sources (cities). However,




as seen in Huszar et al. (2018a), the strong reduction of wind-speed over and around urban areas can be sometimes very strong resulting in counteracting the turbulence decrease. This in turn cause increase of urban PM2.5 concentrations. This probably also contributed to the modelled increases of PM2.5 (e.g. over the Benelux states in JJA, similar to Huszar et al. (2018b)).

Regarding the impact of urbanized land-surface on PM deposition, the results are in-line with the expectations that increased DV over cities (by 30-50%) result in decreased concentrations of PM2.5 (by around -0.12 $\mu gm^{-3}$ and -0.23 $\mu gm^{-3}$ in summer vs. winter). This is a although minor decrease but is seen over the whole domain with maxima, as expected, over cities. Moreover, the decrease of DV is same across all the PM2.5 components as in the used CTM, DV is a function of aerosol size only. Here we have to note however, that the dry-deposition parameterization used here (Zhang et al., 2001) considers urban

areas as flat surfaces with prescribed roughness length and other parameters relevant for the dry-deposition. This certainly differs from reality, where the urban canopy is formed of individual object with different surface materials, and also vegetation fraction while in each of these cases the parameters controlling the dry-deposition are different. E.g. Cherin et al. (2015) showed that "dry deposition velocities can vary by a factor of 24 between two surface types in urban areas". Our results on the impact on urban land-surface on PM2.5 dry deposition are therefor a very rough estimate.

Regarding the impact of modified biogenic emissions on PM2.5, these are of course acting predominantly via modifying SOA concentrations. As BVOC are a main precursor of biogenic SOA, a decrease of biogenic emissions result in decrease of SOA formation. This is more pronounced during summer, of course, when BVOC emissions are at their peak. We also modelled some secondary effects on PNO3 (and to a smaller extent to sulphates and ammonia), which is related to influencing OH radical which in turn influences the oxidation of nitrates and also ammonium causing them to decrease (Aksoyoglu et al.,

2017).

    In further we discuss the modelled RUT-induced modifications of the analysed PM2.5 components. Looking at elemental carbon, it is a inert aerosol in CAMx without chemical decay, so it is influenced via direct pathways along with changes of emissions, meteorological conditions and deposition. Indeed, urbanization increased PEC by about 0.2-0.6 $\mu gm^{-3}$ in winter and by about 0.2-0.3 $\mu gm^{-3}$ in summer, which is mainly influenced by the urban emissions alone causing slightly higher

increases and the increase is predominately limited to urban areas. Indeed, e.g. Skyllakou et al. (2014) showed for Paris, that almost 60% of PEC originates from local sources although they calculated somehow stronger contributions of urban emissions to the total PEC (around 0.3-0.4 $\mu gm^{-3}$). However, Paris is at the high end of the size distribution of cities we selected so its urban emissions will be also very large compared to the average in our selection. The total urbanization impact on PEC is again smaller than that of emission only, which is caused partly due to the effect of UCMF on PEC. Similarly to the total

PM2.5, PEC also responses to higher vertical eddy-diffusion above cities by decreases. A similar decrease of PEC due to UCMF as in our study was modelled in Huszar et al. (2018b) who showed that this decrease is a counteraction of a large decrease due to turbulence enhancement and a smaller increase due to reduced wind-speeds. Our results further showed that the decrease is largest during afternoon hours which is in line with Huszar et al. (2018b) and Huszar et al. (2020a). They shown that the impact of turbulence enhancement is largest during these hours of the day when the strongest mixing in- and

above urban canopy occurs. The impact of enhanced deposition velocities is again expected and exhibits a clear decrease of PEC concentrations above urban areas. The impact is larger in winter in line with the larger winter absolute PEC concentration



compared to summer ones. Further we showed (and this was seen for PM2.5 too) that the strongest decrease of PEC due to increased DV occurs during daytime, which can be clearly explained by the peaking DV values during day and thus strongest influence of urban land-surface (as shown in Fig. 13). This is a know behaviour of particle deposition velocities which are modelled to peak during early afternoon hours (e.g. Nho-Kim et al., 2004). Finally, as PEC is chemically inert, it does not responded to modification of biogenic emissions.

As a result of urbanization, sulphates increased by about 0.05 $\mu gm^{-3}$ in winter and 0.03-0.04 $\mu gm^{-3}$ in summer while urban emissions alone caused a slightly larger increase in winter (around 0.1 $\mu gm^{-3}$) and a similar one in summer. These are almost 10 times smaller values compared to PEC and in general show the low sulphur fraction of urban emission in Europe. Indeed, sulphates are emitted mainly as a result of combustion however strong reduction policies were implemented during the 80s and 90s which substantially reduced the sulphur content of combustion products (Vestreng et al., 2007). Sulphur emissions and thus sulphate formation is larger in eastern Europe, especially in Poland, were coal combustion is still a significant energy producing method. This is reflected also in the emission data we used and consequently, the largest urban contribution to regional sulphate levels are above eastern Europe where often coal combustion facilities are located even within the cities outskirts or coal is used for domestic combustion. For this reason, winter sulphate increases due to urban emission are much larger than during summer. Previously, Skyllakou et al. (2014) showed a very small contribution of local sources to PSO4 concentrations (on the case of Paris) reaching less than 0.1 $\mu gm^{-3}$ during summer, which is in line with the largest contributions in our case (e.g. over Hamburg due to shipping $SO_2$ emissions or over Polish cities). Due to the urban canopy meteorological forcing, sulphates decreased over cities and increased over rural areas. This is expected (similarly to PM2.5) as the main component acting within UCMF is the enhanced vertical eddy diffusion which removes material from the surface model layer and is deposited further from the sources causing increase elsewhere. This behavior was seen also in Huszar et al. (2018b) who have seen an decrease of sulphates of about up to -0.5 $\mu gm^{-3}$, similar to our results. Moreover, sulphates decreased also due to decreases of it precursors, $SO_2$ and $NH_3$ driven by the same mechanism. Indeed, we showed in Huszar et al. (2022) too that $SO_2$ usually decreases above cities by about 0.5 ppbv. $NH_3$ also decreases as seen in Fig. 14 by about up to 1 ppbv in both seasons, limiting the formation of ammonium sulphate. Also Kang et al. (2022) showed that near the surface, sulphates decrease due to the enhanced urban mixing (an increase in the free troposphere above). It has to be noted however, that the UCMF induced modifications are a trade-off between the wind-speed decreases which increase the urban concentrations and the turbulence-induced decreases. In some studies, e.g. Huszar et al. (2018b) the wind speed decrease is more important and can cause that the they have higher urban concentrations due to UCMF. As expected, sulphates decreased as a result of increased dry-deposition velocities and this is amplified also by the increased dry-deposition of $SO_2$ due to urbanization as shown in Huszar et al. (2022). Finally, the impact of reduced urban BVOC emissions on PSO4 is negligible, only some very small increases during summer are modelled. This may be explained by the less OH radical reacting to oxidize biogenic hydrocarbons which thus can oxidize more $SO_2$ (Aksoyoglu et al., 2017).

Urbanization increased nitrates and ammonium by about 0.1 $\mu gm^{-3}$ and 0.25 $\mu gm^{-3}$ respectively with higher numbers in winter, which is clear as the absolute concentration of these secondary aerosols are also higher in winter. The impact of emissions alone is also clearly higher in winter, which is caused by stronger NOx emissions (mainly combustion and transport)



and by usually higher ammonia emissions too during the colder part of the year (although these can have a smaller late summer peak too; e.g. Drugé et al., 2019). Again, the emission impact is higher than the one from total urbanization and this is caused (similar to PM2.5, PEC or sulphates) to the effect of UCMF, which is dominated by increased vertical eddy-diffusion. This

reduces the near surface concentrations of both aerosol as well as of their precursors (NOx and $NH_3$). Previously, Huszar et al. (2018b) showed decreases of PNO3 and PNH4 over central European cities due to the UCMF by about 0.02-0.04 and 0.02 $\mu gm^{-3}$ in summer (they did not looked on winter) which is slightly smaller for nitrates than our numbers and is comparable to ammonium modification presented in this work. The decrease of PNO3 and PNH4 is also caused by the decreases of nitrogen oxides and ammonia as seen on Fig. 14 or previously in Huszar et al. (2018b, 2022). Recently, also Kang et al. (2022)

reported higher nitrate and ammonium values above a large Chinese agglomeration if the urban land-surface and the associated UCMF was not considered. Similar to sulphates, PNO3 and PNH4 responded to urbanization induced land-surface changes by increased dry-deposition velocities resulting in their decrease. In winter, this was partly caused also by the reduced $NH_3$ due to increased dry-deposition, however for JJA, we modelled increased ammonia concentration due to dry-deposition changes due to urbanization (see Fig.14). Indeed, the used dry-deposition model (Zhang et al., 2003) predicts for $NH_3$ in case of dry-

canopies (i.e. summer conditions) smaller dry-deposition velocities for urban areas compared to rural ones (e,g, crops). Finally, as a result of decreased urban BVOC emissions, some increases of nitrates and ammonium is modelled which can be connected to mode OH available to oxidize NOx (Aksoyoglu et al., 2017) but also less NOx is reacting with organic molecules to create organic nitrates (Fischer et al., 2014). E.g. Jiang et al. (2019) showed that smaller biogenic emissions fluxes result in increased PNO3 and PNH4 concentrations over central Europe, while the impact on PNO3 is larger, which is in line with our results.

The impact of urbanization on secondary organic aerosol concentration is notable only during summer owing to the suppressed oxidation of VOC in winter and small BVOC emissions responsible for biogenic SOA formation during the cold part of the year (Gao et al., 2022; Zhai et al., 2023). The total summer impact of urbanization is about 0.03-0.05 $\mu gm^{-3}$ (reaching 0.1 $\mu gm^{-3}$) while the emission impact alone is again larger often exceeding 0.1 $\mu gm^{-3}$. A very similar summer contribution of urban emissions from Paris was modelled earlier by Skyllakou et al. (2014) who pointed out that the contribution of SOA to

the total urban impact is about 5-10%. Freney et al. (2014) arrived to similar results and argued that contribution of SOA gets larger as the urban plume ages. Regarding the impact of UCMF on SOA in summer, it is characterized by both increases and decreases, depending on the city with an average slightly below zero (around -0.01 $\mu gm^{-3}$). The reason for this is probably in the inter-play between the reduction caused by the increased vertical eddy diffusion and increase caused by the decreased urban wind-speeds, while increased urban temperatures shift the partitioning between the gas-phase and particulate-phase to-

wards the gas-phase (Huszar et al., 2018b; Wang et al., 2009). This means the impact over a particular city depends on relative magnitude of these three components of UCMF and how they act on SOA. The UCMF caused also a relatively large impact over rural areas, which can be explained also by the fact that SOA is formed more readily in an aged urban plume Ortega et al. (2016), so probably the removed SOA precursors by the increased vertical eddy diffusion were transported over rural areas while oxidized into SOA. Regarding the impact of increased PM deposition velocities due to urbanization, the impact follows

those for secondary inorganic aerosol or PEC, i.e. the urban concentrations decreased (by around -0.02 $\mu gm^{-3}$). Here again, one must realize that the the modified DVs impacted also the precursor species which oxidize to semi-volatile compounds. In



CAMx in the Zhang deposition model for gases (Zhang et al., 2003), the deposition velocities for SOA precursors (and most of VOCs) are smaller for urban canopies which means their concentration are larger for urban areas. Hence more SOA formation is favored because higher precursors abundances. This consequently means that the final impact on SOA is a counteract

between the decrease due to direct impact of urbanization land-use change on SOA deposition velocities and increase due to higher precursor concentrations. In our simulation the former was the dominating effect as SOA decreased above all urban areas.

Finally, the urbanization induced decrease of BVOC emissions resulted in a reduction of SOA concentrations (by about 0.04-0.06 $\mu gm^{-3}$) and this decrease is modelled not only over urban areas but over large rural regions while the largest decrease is

635 above Milan which has also the warmest climate among the selected cities. Indeed, SOA of biogenic origin is a an important contributor to urban SOA levels so the modelled decreases are were expected (Couvidat et al., 2013; Hu et al., 2017). Ghirardo et al. (2016) also calculated strong influence of local BVOC emissions from urban trees in a Chinese megacity (although the anthropogenic influences were much larger). Again, the impact affected also rural areas which can be explained by the repeated fact that in aged urban plumes, so forms more effectively.

In summary, we evaluated over an ensemble of 19 Central European cities the impact of rural-to-urban transformation and its four contributors on PM urban (and rural) concentrations including the impact on its primary and secondary components. We found that the two main controlling drivers are the impact of urban emissions themselves (increase the concentrations for PM2.5 and all of its analyzed components) and the urban canopy meteorological forcing (usually decreases over urban areas, increases over rural ones). We showed however that two additional controlling mechanisms can play important role within the

process of urbanization although smaller by an order of magnitude than the effect of emissions and UCMF: the impact of dry-deposition velocity changes due to urbanization of land-surface and the reduction of biogenic emissions by turning rural/natural land-surface into urban built-up. The former results in decreases of concentrations due to increased deposition velocities, the latter act predominantly via modification of secondary organic aerosols and results in decrease of PM2.5 concentrations (by reducing SOA). In summary, when the impact of urbanization on PM air-pollution is analyzed, all four contributors have to be

accounted for.

We must also stress that that the cities selected in this study are from a relatively small region meaning that they do not exhibit a substantially different background climate. Moreover the typical "rural" vegetation was assumed to be crop which might not be the case if cities over other parts of the world were considered (e.g. tropical areas) meaning the that the impact of modified biogenic emissions could be much stronger. Further, some secondary effect of PM concentration changes can play role too via

the direct and indirect radiative effect. E.g. photolysis rates and temperatures are altered via the direct effect of aerosol that in turn influences air chemistry (Han et al., 2020; Wang et al., 2022) or the vertical structure of urban boundary layer can be altered by the aerosol emitted that modifies the overall stability and convection (Miao et al., 2020; Slater et al., 2022; Fan et al., 2020; Yu et al., 2020; López-Romero et al., 2021) which in turn can modify the vertical mixing and precipitation with feedbacks on species concentration. Consequently, to obtain a more accurate quantification of the impact of rural-to-urban

transformation on PM, these effect has to be included in modelling studies.



*Code and data availability.* The RegCM4.7 model is freely available for public use at https://github.com/ICTP/RegCM (last access May 16, 2023) (Giuliani, 2021). CAMx version 7.10 is available at http://www.camx.com/download/default.aspx (last access May 16, 2023) (Ramboll, 2020).The RegCM2CAMx meteorological preprocessor used to convert RegCM outputs to CAMx inputs and the MEGAN v2.10 code as used by the authors is available upon request from the main author. The complete model configuration and all the simulated data

(3-dimensional hourly data) used for the analysis are stored at the Dept. of Atmospheric Physics of the Charles University data storage facilities (about 5TB) and are available upon request from the main author.

*Author contributions.* PH created the concept and designed the experiments, PH and JK performed the model simulations, APPP, LB and AVP contributed to input data preparation, model configuration and analysis of the outputs, ALL authors contributed to the manuscript text.

*Competing interests.* The authors declare that they have no conflict of interest.

*Acknowledgements.* This work has been funded by the Czech Science Foundation (GACR) project No. 19-10747Y, by the Czech Technological Agency (TACR) grant No.SS02030031 ARAMIS (Air Quality Research Assessment and Monitoring Integrated System) and by project – Programmes of Charles University. We further acknowledge the CAMS-REG-APv1.1 emissions dataset provided by the Copernicus Atmosphere Monitoring Service, the Air Pollution Sources Register (REZZO) dataset provided by the Czech Hydrometeorological Institute and the ATEM Traffic Emissions dataset provided by ATEM (Studio of ecological models). We also acknowledge the providers of AirBase

European Air Quality data (http://www.eea.europa.eu/data-and-maps/data/aqereporting-1).



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



**Table 1.** The list of CAMx simulations performed with the information of the effects considered.

| | Experiment | Regional Chemistry Transport Model (CAMx) simulations | | | |
|---|---|---|---|---|---|
| | | Driving meteorology | Urban emissions | Land-use (deposition) | BVOC emissions |
| 1 | ENNNN (Reference) | Nourban | No | Nourban | Nourban |
| 2 | ENYNN | Nourban | Yes | Nourban | Nourban |
| 3 | ENYUN | Nourban | Yes | Urban | Nourban |
| 4 | ENYUU | Nourban | Yes | Urban | Urban |
| 5 | EUYUU | Urban | Yes | Urban | Urban |





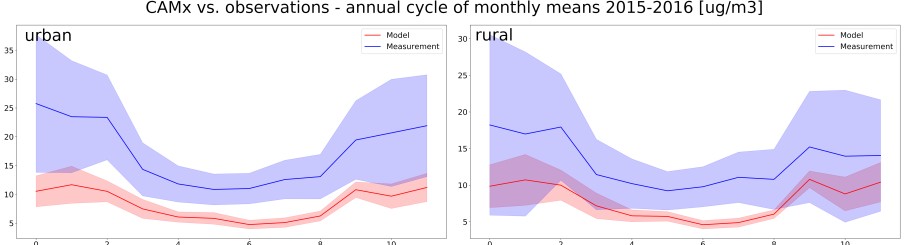

**Figure 1.** Comparison of modelled (red) and observed (blue) PM2.5 monthly concentrations over rural (left) and urban (right) AirBase stations as 2015-2016 mean. Shaded areas represent the standard deviation across all the stations. Units are in $\mu\mathrm{gm}^{-3}$.

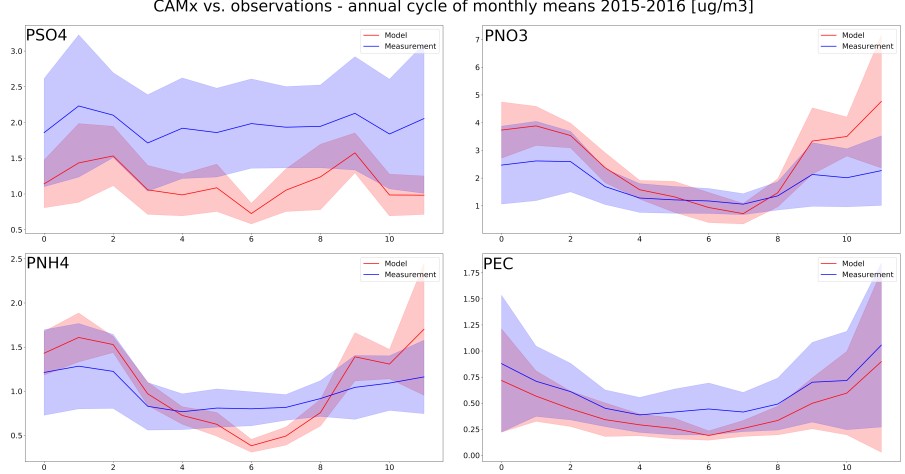

**Figure 2.** Comparison of modelled (red) and observed (blue) PSO4, PNO3, PNH4 and PEC monthly concentrations over available EMEP stations as 2015-2016 mean. Shaded areas represent the standard deviation across all the stations. Units are in $\mu\mathrm{gm}^{-3}$.



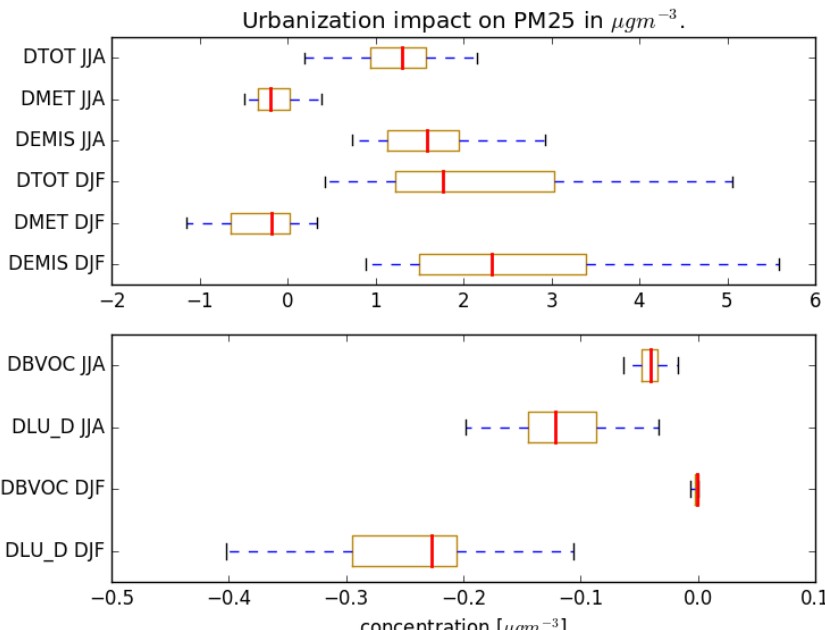

**Figure 3.** The 2015-2016 DJF and JJA averaged total impact of urbanization as well as of each contributor to the urban concentrations of PM2.5 averaged over all chosen city. The boxplots show the 25% to 75% quantiles including the minimum and maximum value across all cities. The red line shows the median value. Values are taken from model grid-cell that covers the city center. The upper sub-figures show the two main contributors including the total impact ("DEMIS", "DMET" and "DTOT") while the lower one the minor contributors ("DLU_D" and "DBVOC"). Units are in $\mu gm^{-3}$.



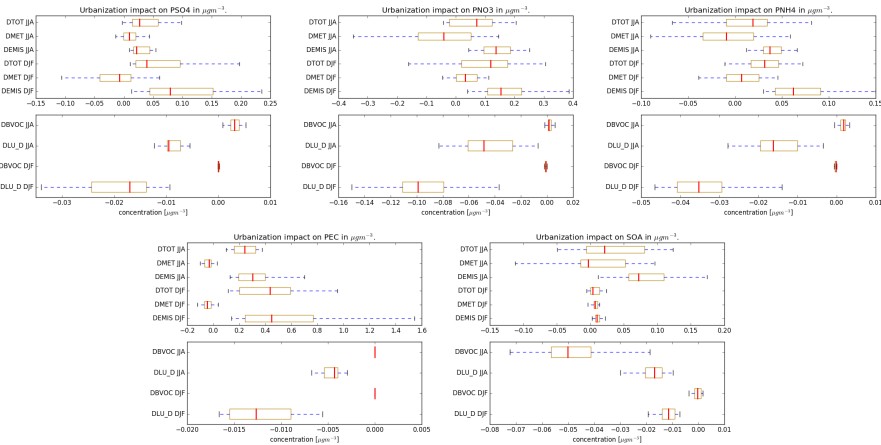

**Figure 4.** Same as Fig. 3 but for PM2.5 components: PSO4, PNO3, PNH4 (upper row), PEC and SOA (lower row).



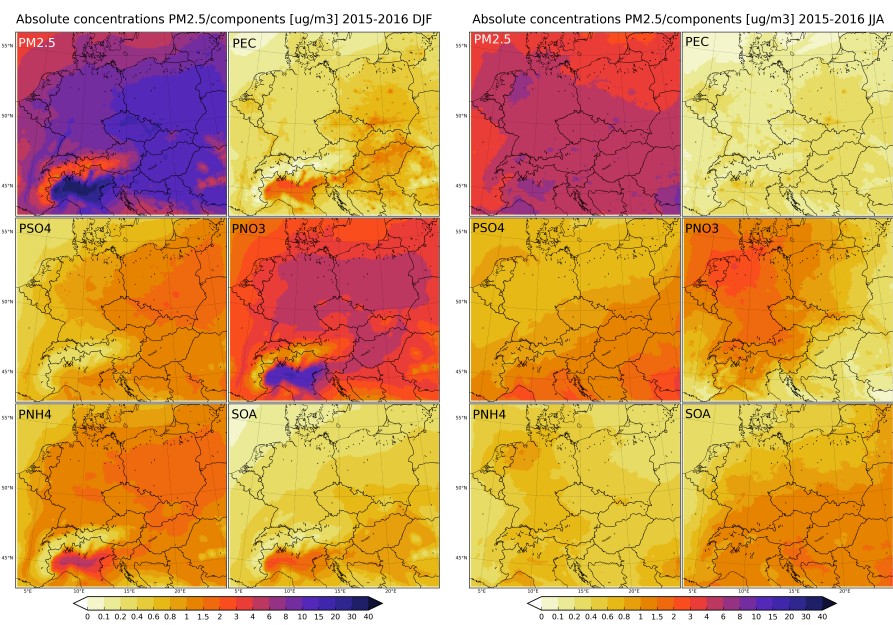

**Figure 5.** The absolute DJF (left panel) and JJA (right panel) concentrations of PM2.5 and its components (PEC, PSO4, PNO3, PNH4 and SOA) averaged over 2015-2016 in $\mu\mathrm{gm}^{-3}$.



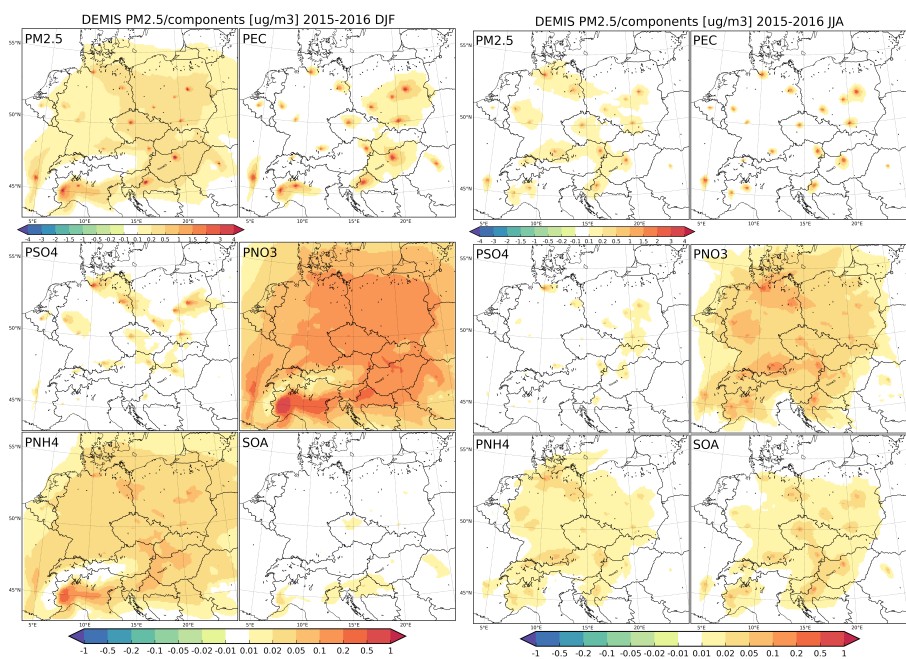

**Figure 6.** The spatial distribution of the 2015-2016 DJF (left panel) and JJA (right panel) average urban emission impact "DEMIS" on PM2.5 and its components. Units in $\mu g m^{-3}$. Note that PM2.5 has separate colorbar.



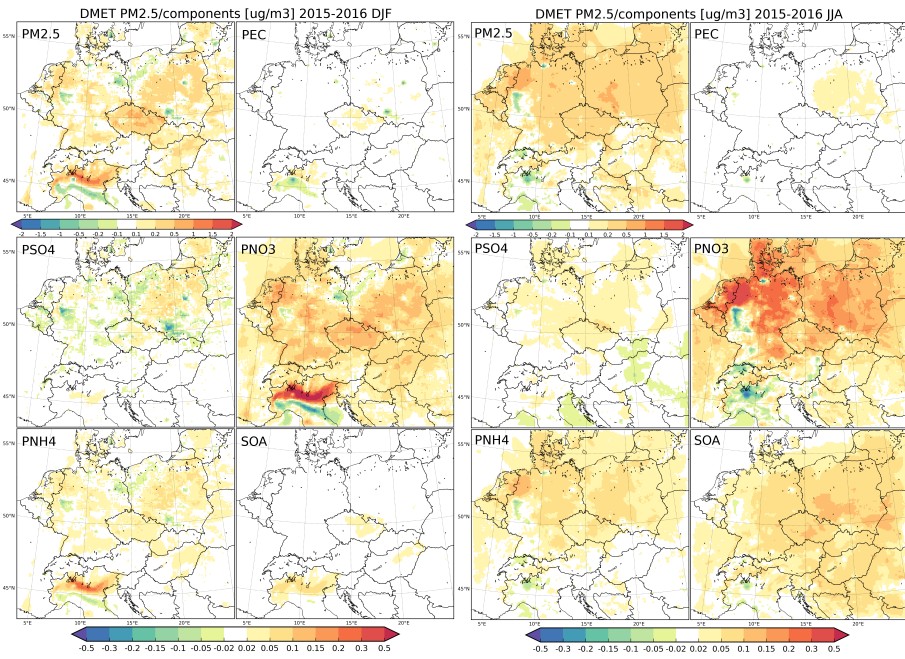

**Figure 7.** The spatial distribution of the 2015-2016 DJF (left panel) and JJA (right panel) average UCMF impact "DMET" on PM2.5 and its components. Units in $\mu\mathrm{gm}^{-3}$. Note that PM2.5 has separate colorbar.





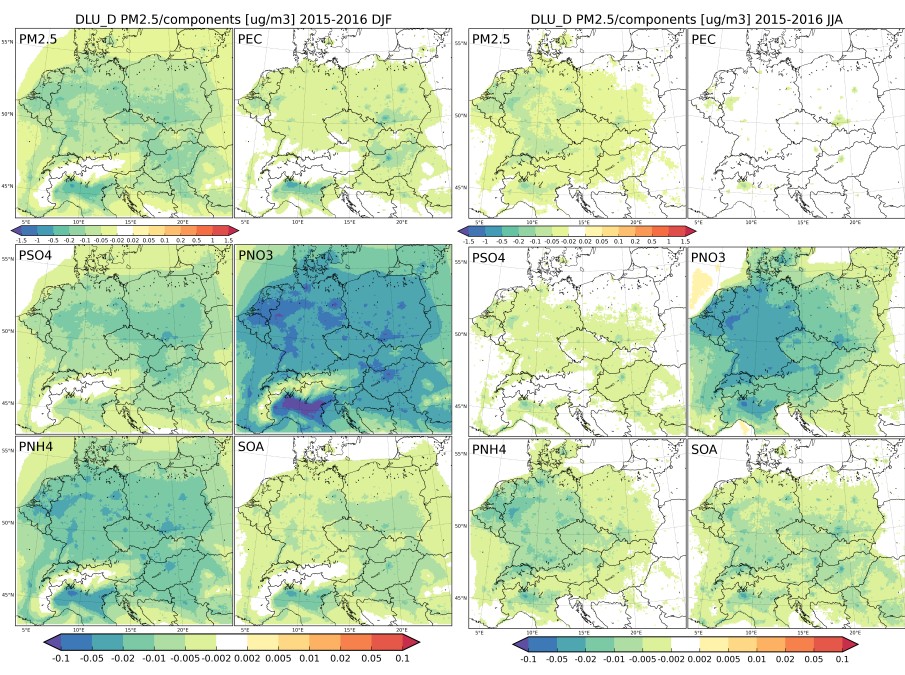

**Figure 8.** The spatial distribution of the 2015-2016 DJF (left panel) and JJA (right panel) average impact of modified dry-deposition velocities due to land-use change "DLU_D" on PM2.5 and its components. Units in $\mu g m^{-3}$. Note that PM2.5 has separate colorbar.

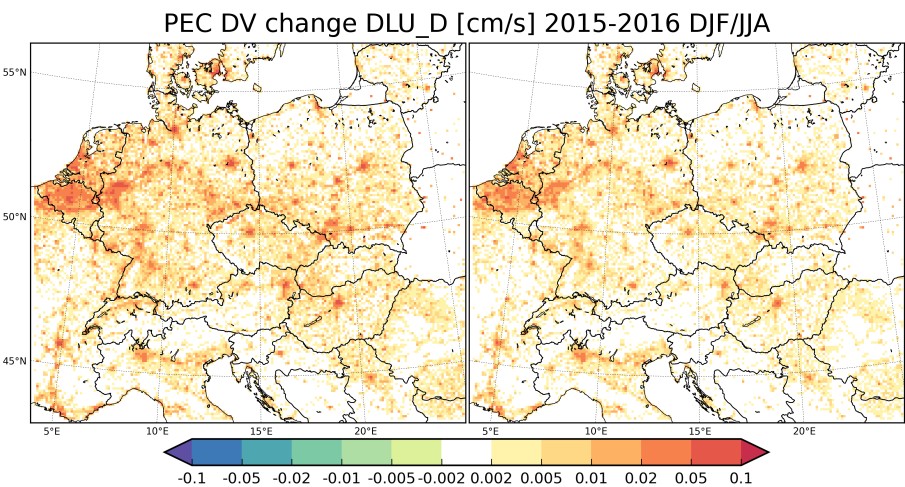

**Figure 9.** The spatial distribution of the 2015-2016 DJF (left panel) and JJA (right panel) average impact of urban land-use on dry-deposition velocities for PEC. Units in $\mathrm{cms}^{-1}$.





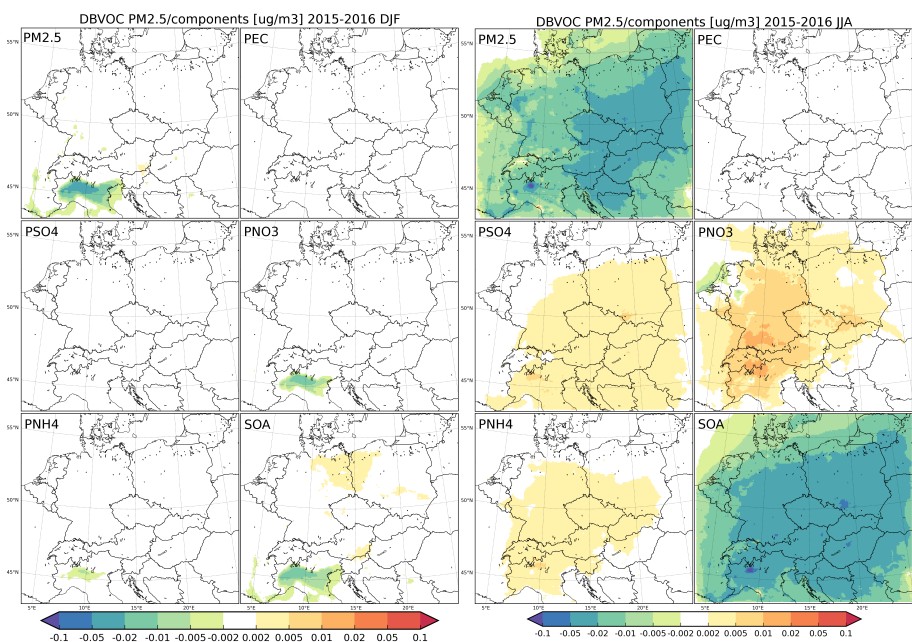

**Figure 10.** The spatial distribution of the 2015-2016 DJF (left panel) and JJA (right panel) average impact of modified biogenic emissions "DBVOC" on PM2.5 and its components. Units in $\mu$gm$^{-3}$.

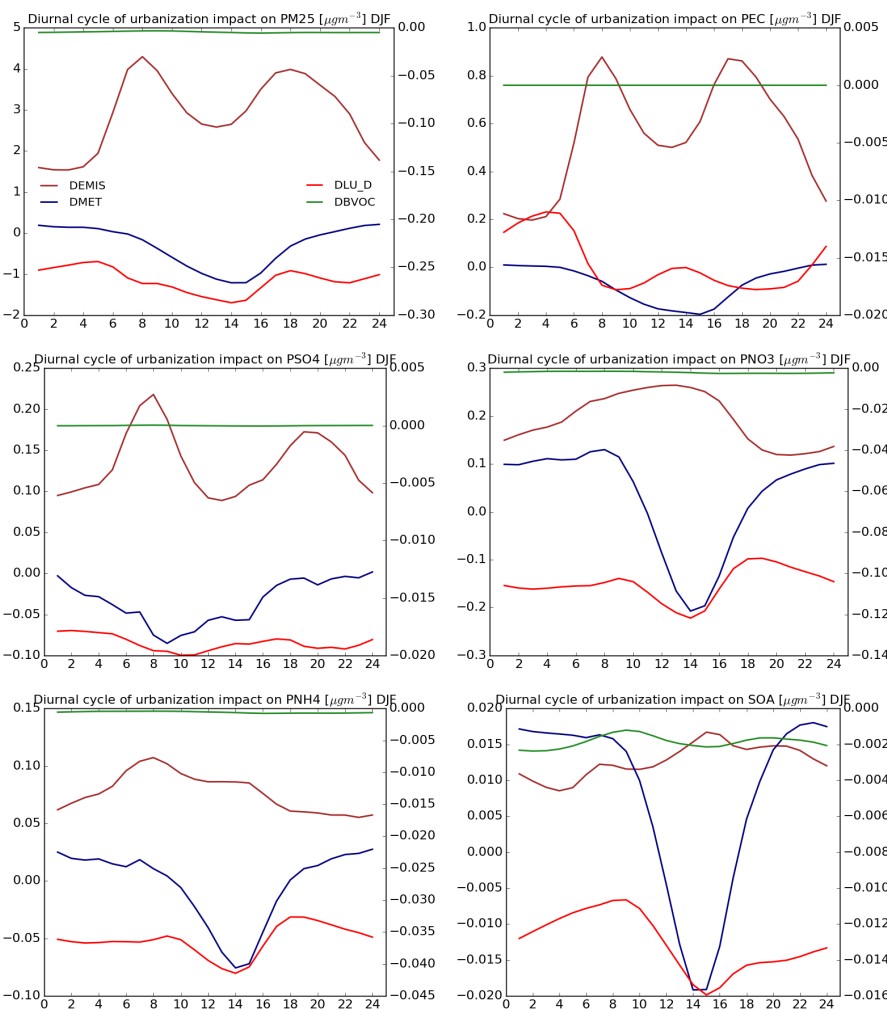

**Figure 11.** Diurnal cycles of the impact of individual contributors to RUT averaged over 2015-2016 DJF for PM2.5, PEC, PSO4, PNO3, PNH4 and SOA. Colors stand for: brown – DEMIS, blue – DMET, red – DLU_D and green – DBVOC. Left y-axis is for the two major contributors: DEMIS and DMET, while the right y-axis belongs to the two smaller contributors (DLU_D and DBVOC). Units are in $\mu gm^{-3}$.





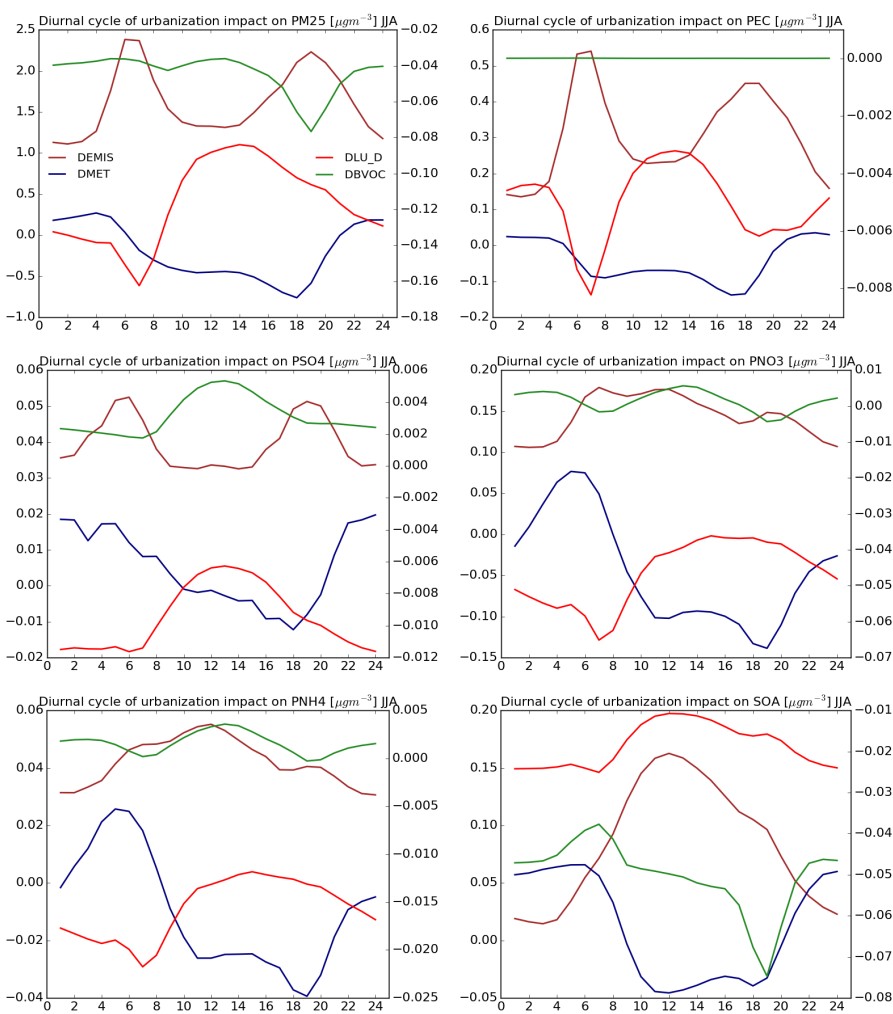

**Figure 12.** Same as Fig 11 but for JJA.





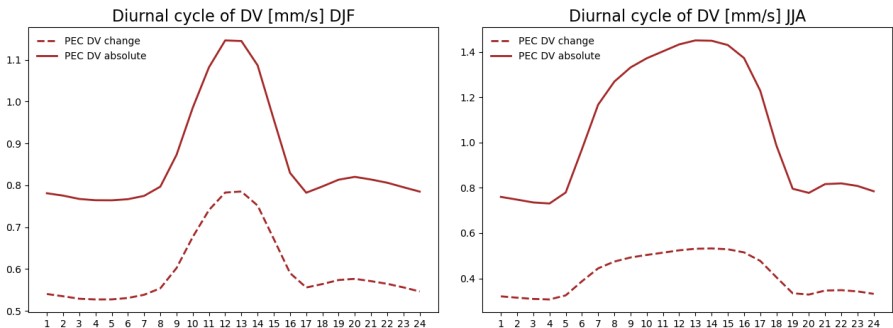

**Figure 13.** Diurnal cycle of the DJF (left) and JJA (right) DV of PEC: solid lines denote absolute values, dashed lines mean the change caused by the urban land-surface. Units in $\mathrm{mms^{-1}}$.

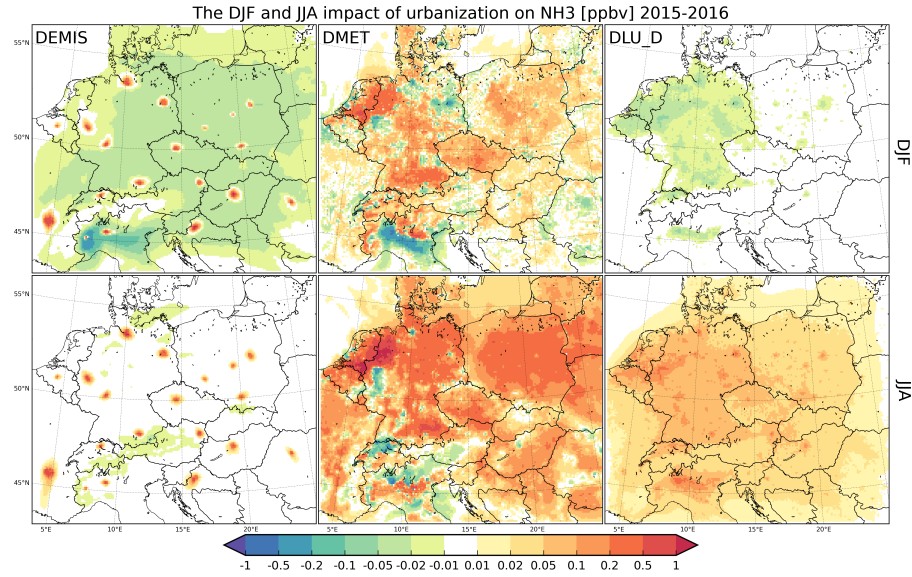

**Figure 14.** The DJF (upper row) and JJA (lower row) impact of "DEMIS", "DMET" and "DL_U" on near surface $NH_3$ concentrations in ppbv.