# Peer review of "Impact of urbanization on fine particulate matter concentrations over central Europe"

_EGUsphere, 2023_

## Author Comment (AC1)

**Authors response on the Anonymous Referee #1 review of "Impact of urbanization on fine particulate matter concentrations over central Europe"**

by Huszar et al. (acp-2023-1037)

Dear Anonymous Referee #1,

thank you for your time and effort to review our paper and for all your comments. Please find our point-by-point answers to the points of your revision (in bold italic) below.

*This manuscript discusses the effect of urbanization on PM 2.5 levels in central Europe. Four factors were considered, i.e., the urban emissions, the urban canopy meteorological forcing, the impact of modified dry deposition velocities and the impact of modified biogenic emissions. A cascade of simulations with RegCM and CAMx were performed to separate the effect of these factors. The authors found that urban emission contributes the most to increasing the PM 2.5 concentration, while urban canopy has a counter effect due to stronger vertical eddy-diffusions. The transformation of land-use also tends to decrease PM 2.5 levels by increasing the particle dry deposition velocity. Overall, the conclusions drawn from the simulations seem sound, but I believe the presentation of the results can be improved and the manuscript can become more concise by revision.*

*Major comments:*

- *Regarding the design of the simulation cascade, more explanation of the simulation sequence of is needed. If the simulations follow a different sequence, will the results change significantly?*

Authors response: Each of the four contributors (impacts), in general, depends on the "base" state to which it was added to. Therefor we aimed to ensure that it is as close to reality as possible. Consequently, we started with addition of the urban emission ("DEMIS") to the reference (non-urbanized) state as this impact was assumed (and proven) to be the dominant. With this choice, the urban atmosphere was already filled with pollutants serving a base state for the other impacts. The effect of the order of addition of the two further contributors, DLU and DBVOC is probably small as their magnitude is also small. Here the only thing that matters is whether they are applied before or after DEMIS. Doing so before DEMIS would lead to even smaller DLU and DBVOC impacts. Further, in our previous paper (Huszar et al., 2022), which had the same goal and modeling design as this paper but looked at the gas-phase chemistry, we analyzed in detail the effect of the order of the two sub-contributors of DBVOC, which are the impact of reduced vegetation (DBVOC_L; see Huszar et al, 2022) and impact of changed meteorological conditions that drive the MEGAN model impacting BVOC fluxes (DBVOC_M). We found the 1) with regard to DBVOC, the changes of vegetation cover play a much more important role than the changed meteorological conditions 2) the partial impact of changed meteorological conditions is smaller if applied after DBVOC_L.

As for the DMET impact, this is in general the second strongest contributor and one gets different magnitudes of the impacts if DMET is applied before DEMIS. In other words the order of the two strongest contributors matters a lot. Let's suppose that first the DMET contributor has been applied. This means that the meteorological conditions that drive the impact of emissions already include the urban canopy meteorological forcing (UCMF). Huszar et al. (2021) however showed that the impact of urban emissions is considerably (almost by 50%) smaller if UCMF is considered. On the other hand DMET when applied as the first contributors would be very small as UCMF would act on much less polluted air over cities (with missing urban emissions).

So in conclusion, the DLU, DBVOC and DMET impact would be smaller if applied before DEMIS and somehow decoupled from the reality motivating us to start the addition of the different components by the emissions themselves. The choice of the order of the other three contributors has rather much smaller effect.

In the revised manuscript, we dedicated paragraph in the Discussion to discuss these issues.

*•The authors have somewhat strictly separated results from discussions. I find this way of writing very difficult to follow. One the one hand, in the results section some paragraphs are only recounting figure contents and barely provide any explanations (e.g., sections 3.3.1-3.3.4). This makes the reading process really dry (actually, a table with numbers may actually outperform the texts in 3.3.1-3.3.4). One other hand, when I arrive at the discussion part (far away from the results), I have a hard time correlating the discussions to the corresponding results; as a result, I have difficulty in assessing the validity of some statements in the discussion section. Therefore, I recommend restructuring of the paper to some extent by combining some of the discussions with the results.*

Authors response: We admit that a sharp separation of the results and any related discussion makes the reading of the manuscript and understanding/interpretation of the results difficult. We however would lean to this tradition and somehow keep these two parts of the manuscript separated (as we did in our previous related studies too).

However, as a compromise, we included in the Results section – at the end of individual subsections and/or paragraphs a small interpretative/explanatory text to put the presented results into some context while referring to the Discussion section for a more comprehensive discussion. Moreover, to facilitate the reading of the Discussion, we included the Figure references also in the Discussion text.

**Minor comments:**

*•Line 175: Eqn (3) assumes that the processes are additive. This sentence seems to suggest the opposite causal relation.*

Authors response: We assume that the referee meant Line 275. yes, we agree that this sentence suggest the opposite causal relation. We meant here that the impacts are simultaneous and act together, and, if they were calculated by removing them one-by-one always from the full experiment ("EUYUU") then they would not be additive, i.e. the total impact is not the sum of the individual impact. In our case, the additivity is ensured only by the fact, that they are added one-by-one to the reference case, i.e. as Eq. 3 suggests.  This is now clearly explained in the manuscript to avoid confusion about the (non)additivity.

*•Figure 5: Please state clearly whether figure 5 shows the measured or simulated concentrations.*

Authors response: These are modelled concentrations from the experiment 5 ("EUYUU") which considers all urban effects. This fact has been added to the corresponding paragraph and also to the figure's caption.

**Technical:**

**There are quite a number of tiny mistakes in the manuscript. The authors need to carefully go through the manuscript for a more complete check.**

Authors response: we carefully re-read the text and corrected these mistakes.

**For the majority of the figures: Please use larger font size for the axis tick labels. Figures 1, 2, 11, 12 also need proper labels for the axes. In Figures 11 and 12, the phrase 'Diurnal cycle of urbanization impact' does not need to appear above every panel.**

Authors response: We increased the axis tick labels where they were too small (mainly for the 2D shaded plots) and labeled the axes for the temporal plots (annual and diurnal cycles). We also removed the multiple repeated subtitles and used only one common for all the sub-figures.

**Line 32: the abbreviation of DV has not been defined previously.**

Authors response: the abbreviation has been defined in the abstract.

**Line 65: messed up citation of Yang et al.**

Authors response: Fixed.

**Line 83: this sentence needs to be revised.**

Authors response: The sentence has been slightly simplified and rephrased to be more comprehensible.

**Line 163: This sentence is very confusing.**

Authors response: The last two sentences of the paragraph were rephrased to be more clear and logical.

**Line 237: although -> despite**

Authors response: Changed.

**Line 282: the impacts of urbanization on POA.**

Authors response: Corrected.

**Line 285: … and their components to observations.**

Authors response: Corrected.

**Line 291: and the underestimation is stronger in winter.**

Authors response: Modified.

**Line 310: missing units**

Authors response: Units added (throughout the whole manuscript where missing)

**ug/m3 are sometimes italic and sometimes normal.**

Authors response: Units changed normal in all cases (except some figures where the plot software used puts italic font for units containing Greek letters by default).

**Line 541: we further discuss…**

Authors response:

**Line 559: know -> known**

Authors response: Corrected.

**Line 561: responded -> respond**

Authors response: Corrected.

**Line 623: the citation of Ortega et al. is in the wrong format**

Authors response: Corrected.

References:

Huszar, P., Karlický, J., Marková, J., Nováková, T., Liaskoni, M., and Bartík, L.: The regional impact of urban emissions on air quality in Europe: the role of the urban canopy effects, Atmos. Chem. Phys., 21, 14309–14332, https://doi.org/10.5194/acp-21-14309-2021, 2021.

Huszar, P., Karlický, J., Bartík, L., Liaskoni, M., Prieto Perez, A. P., and Šindelářová, K.: Impact of urbanization on gas-phase pollutant concentrations: a regional-scale, model-based analysis of the contributing factors, Atmos. Chem. Phys., 22, 12647-12674, https://doi.org/10.5194/acp-22-12647-2022, 2022.

**Authors response on the Anonymous Referee #2 review of "Impact of urbanization on fine particulate matter concentrations over central Europe"**

by Huszar et al. (acp-2023-1037)

Dear Anonymous Referee #2,

thank you for your time and effort to review our paper and for all your comments. Please find our point-by-point answers to the points of your revision (in bold italic) below.

*Summary: In the manuscript titled "Impact of urbanization on fine particulate matter concentrations over central Europe", the authors examine urban contribution to particulate matter and its constituents over central Europe using a modeling approach. The study is overall written well and mostly well designed. I do have several questions about the modeling approach and uncertainties and how they might impact the results.*

*Major comments:*

*1.  In general, I found the modeling approach difficult to follow. There seem to be multiple models driving each other and other models being used to derive or modify the emission inventories (for instance, MEGAN for BVOC emissions). It would really help to have a schematic of the overall modeling setup and how each component feeds into each other regarding input and output.*

Authors response: We admit that such a figure would facilitate the understanding of the different models used including their mutual interaction via data flow. For this goal, we compiled a new figure (Fig.1.) showing the three main model families used (for meteorology, for air-quality and for emissions) and their components, which are the online coupled surface/urban model (CLM4.5/CLMU) within RegCM, the biogenic and athropogenic emissions models (MEGAN and FUME) within the overall emissions processing. The figure also shows which data are exchanged between the models.

*2.   As far as I can tell, the RegCM model uses the urban representation of CLM4.5, which is a single layer canopy model with no urban vegetation and no anthropogenic heat flux from vehicles. How does this affect the BVOC estimates. Does the MEGAN algorithm actually resolved urban vegetation and biogenic emissions from them (or deposition on them?). Additionally, where did the leaf area index and plant functional type data for urban areas come from? Urban areas can have very different vegetation characteristics from their surrounding rural areas (Paschalis et al. 2021).*

Authors response: Indeed, RegCM uses the CLMU urban canopy model within the CLM4.5 land model for "urban" landunits. These do not include any vegetation, however cities are covered also by vegetation (e.g. parks) and these are included in the "vegetated" landunit in CLM4.5. The fractional landuse representation within RegCM/CLM4.5 allows to account for even the smallest fractions of vegetation. The emissions from this vegetation depend on two factors: plant-functional-type (PFT, constant in time) and leaf-are-index (LAI; varying in time). These data are provided to MEGAN as inputs (see Sindelarova et al. 2014). LAI are taken from MODIS satellite sampling so might represent the real situation reasonably.  As for PFT,  these are taken from Lawrence and Chase (2007). In this dataset, PFT considers the large variety and spatial differences of vegetation including that over cities. Moreover, PFT data are represented as fraction of different functional types within the vegetated surface fraction, so (again) even the smallest fraction of a plant type within a vegetated landunit in a city is accounted for. We admit that the PFT data can be inaccurate for some cities or neighborhoods, as for the chosen cities, some urban development took place between 2007 and 2023, but the majority of the urban design remained the same. This is noted in the revised manuscript.

As the fractional manner of the representation of vegetation holds also for CAMx, and the used deposition model in CAMx (Zhang) considers deposition on plants, this means that deposition on urban vegetation is accounted for (in a same way as the deposition on any other vegetation).

As for the heat flux from vehicles, this is not considered in our simulation so we can assume that urban temperatures might be somewhat underestimated due to this missing heat source.

*3. I have other questions about the emission inventory. How are vehicular emissions considered in these estimates? What about impacts from asphalt-related emissions (Khare et al. 2020)?*

Authors response: The traffic combustion related emissions are of course part of the CAMS as well as Czech emission data (REZZO and ATEM). The re-suspension of dust from traffic is included in the Czech emission fluxes but is not included in the European CAMS data. As for the asphalt related emissions, we admit that these can be an important contributor to SOA precursors, but in the current version of the anthropogenic emission model they are not included. We made these facts clear in the manuscript ("Model setup and data" section) along with the potential impact on the results (which is basically underestimation of the SOA loads and overall fine PM)

*4. Finally, as I understand it, these are offline simulations. Does that mean that there is no feedback from the land to the atmosphere? In the real world, urban heat and moisture islands (Chakraborty et al. 2022) may impact near-surface chemistry and convection; and thus overall pollutant concentrations within and sourced from urban areas. There needs to be a broader discussion about these uncertainties.*

Authors response: The coupling between the chemistry transport model (CAMx) and the regional climate model (RegCM) is offline, however, the urban canopy model inherited within the surface model (CLM4.5) is online coupled to RegCM (or more precisely, it is part of). This means that all the urban effects, which include also those listed by the referee, i.e. urban heat island, moisture island (and others, like increased vertical turbulence, lower over wind speeds) are included in the meteorological data that is offline fed into CAMx and into MEGAN. Thus these effect indeed have effect on chemistry in our simulations. We made this fact more clear in the revised manuscript.

*Minor comments:*

*1. Figs 1 and 2: Please add the y axis labels with the units.*

Authors response: Axis labels added (and tick-fonts enlarged – following the other referee's comment)

*2. Page 1: 'We evaluated the RUT impact on PM2.5 over an ensemble of 19 central European cities': That is not the typical usage of ensemble. 19 is just the sample size here?*

Authors response:  Indeed, 19 here is the sample size. We thus replaced "ensemble" to "sample".

*References:*

Lawrence, P. J. and Chase, T. N.: Representing a new MODIS consistent land surface in the Community Land Model (CLM 3.0), J. Geophys. Res.-Biogeo., 112, G01023, https://doi.org/10.1029/2006JG000168, 2007.

*Paschalis, A., Chakraborty, T. C., Fatichi, S., Meili, N., & Manoli, G. (2021). Urban forests as main regulator of the evaporative cooling effect in cities. AGU Advances, 2(2), e2020AV000303.*

Sindelarova, K., Granier, C., Bouarar, I., Guenther, A., Tilmes, S., Stavrakou, T., Müller, J.-F., Kuhn, U., Stefani, P., and Knorr, W.: Global data set of biogenic VOC emissions calculated by the MEGAN model over the last 30 years, Atmos. Chem. Phys., 14, 9317--9341, https://doi.org/10.5194/acp-14-9317-2014, 2014.

Khare, P., Machesky, J., Soto, R., He, M., Presto, A. A., & Gentner, D. R. (2020). Asphalt-related emissions are a major missing nontraditional source of secondary organic aerosol precursors. Science advances, 6(36), eabb9785.

Chakraborty, T., Venter, Z. S., Qian, Y., & Lee, X. (2022). Lower urban humidity moderates outdoor heat stress. Agu Advances, 3(5), e2022AV000729